# Modeling and verifying ice supersaturated regions in the ARPEGE model for persistent contrail forecast

Sara Arriolabengoa[1], Pierre Crispel[1], Olivier Jaron[1], Yves Bouteloup[1], Benoît Vié[1], Yun Li[2], Andreas Petzold[2], and Matthieu Plu[1]

[1]Météo-France, CNRS, Univ. Toulouse, CNRM, Toulouse, France
[2]Institute of Climate and Energy Systems: Troposphere (ICE-3), Forschungszentrum Jülich, Jülich, Germany

**Correspondence:** Sara Arriolabengoa (sara.arriolabengoa@meteo.fr) and Pierre Crispel (pierre.crispel@meteo.fr)

**Abstract.**

Contrails formed by aircraft in ice supersaturated regions (ISSR) can persist and spread for several hours, evolving into cirrus which have a net positive effect on global warming. Reducing this contribution could be achieved through on-purpose flight planning, in particular by avoiding ice supersaturated regions. In this context, a modification to the cloud scheme of the ARPEGE (Action de Recherche Petite Echelle Grande Echelle) operational numerical weather prediction (NWP) model is proposed to enable the representation of ISSRs at cruise altitude. This modification does not require any major algorithmic changes or additional computational effort, and the methodology is transferable to similar parameterizations, commonly used in global circulation models.

Humidity forecasts are evaluated using in situ aircraft humidity observations and compared with operational forecasts from ARPEGE and the Integrated Forecast System (IFS). A sensitivity study on neighborhood tolerance and humidity thresholding is carried out, enabling a comprehensive comparison between NWP forecasts. It is shown that the modified cloud scheme allows for supersaturation, significantly improving the representation of humidity with respect to ice, with ISSR discrimination skills close to IFS (hit rate $\sim$ 80 % and false alarm ratio $\sim$ 30 % when a neighborhood tolerance of 150 km, i.e. 10 min of flight, is applied). The spatial correspondence between observations and the modified ARPEGE model is illustrated by a commercial flight case study. The modelization of ice supersaturation in ARPEGE can therefore be used for further contrail climate impact applications, together with the associated evaluation methodology, which contributes to the definition of a shared framework for ISSR verification.

## 1 Introduction

Contrails, also known as condensation trails, are high-altitude ice clouds that form when water vapor from fuel combustion is released by jet engines. When temperatures are low enough and exceed the so-called Schmidt-Appelman criterion (Schumann, 1996), engine emissions lead to the formation of straight cirrus clouds, which are often clearly visible behind aircraft. When triggered in ice supersaturated regions commonly denoted ISSRs (i.e. relative humidity w.r.t ice $RH_{ice} > 100$ %), the trail can

spread to form a large cirrus cloud, which can persist for several hours and is clearly visible by satellite (Minnis et al., 1998; Gierens and Vázquez-Navarro, 2018).

According to various studies (Lee et al., 2009, 2021), and in view of increasing civil air traffic, these induced cirrus clouds are likely to contribute significantly and positively to global warming. The effective radiative forcing (ERF) associated to persistent contrail could be of the same order of magnitude as ERF issued from $CO_2$ emissions of aviation industry, even if large uncertainties remain, stemming from various sources, such as our incomplete understanding of contrail cirrus life cycle and radiative properties (Schumann and Heymsfield, 2017).

The impact of persistent contrails on global warming might be mitigated by optimizing pre-tactical flight paths or by tactical avoidance of regions where meteorological conditions are likely to be favorable to persistent contrails (Sausen et al., 2023). Studies by Teoh et al. (2020) suggested that only 2 % of flights could contribute to 80 % of the estimated contrail ERF by using algorithmic climate change function (van Manen and Grewe, 2019). Other studies by Matthes et al. (2020) involving climate-optimization simulations found that the aviation-induced climate impact can be reduced in the range of 50 % at the expense

of 1-2 % additional operational costs. The potential mitigation options at pre-tactical and tactical time scale to reduce climate impacts require operational weather forecasts, preferably at continental or global scale. These models must be reliable to predict where contrails may form and persist, which requires sufficient quality on temperature and relative humidity estimates in the upper troposphere and lower stratosphere (UTLS) region.

    Most global circulation models (GCM) do not properly represent humidity in these regions because the parametrization of

cirrus cloud formation implements saturation adjustment. One of the first operational parametrizations of supersaturation w.r.t ice in a global NWP model was proposed by Tompkins et al. (2007) and implemented in the Integrated Forecast System (IFS) of the European Center for Medium-Range Weather Forecasts (ECMWF). Ice supersaturation is parametrized within the cloud scheme by modifying the amount of water vapor in the clear-sky portion of the grid-box according to temperature conditions. However, this scheme is associated with a prognostic cloud fraction description which is non-easily transferable to many other

global circulation models where cloud fraction is diagnosed. In general, obtaining a reliable description of the ISSR remains a challenge (e.g. Gierens et al., 2020) and very recent works are being done on modified micro-physical schemes to allow or enhance the description of ISSR (e.g. Sperber and Gierens, 2023; Borella et al., 2024; Seifert, 2024; Thompson et al., 2024), making this topic a very active field of research. In the global NWP ARPEGE (Action de Recherche Petite Echelle Grande Echelle) model of the French national weather service Météo-France (Bouyssel et al., 2022), cloud fraction is diagnosed and

the current saturation adjustment scheme converts instantaneously any excess of water vapor above ice saturation into ice, which leads to the result that ISSR cannot be represented.

    The growing interest for ISSR forecasting for operational mitigation strategies should also trigger appropriate verification methods of NWP models. Most verification studies (e.g. Tompkins et al., 2007; Reutter et al., 2020; Gierens et al., 2020; Sperber and Gierens, 2023) compared the distribution of temperature and relative humidity to in situ measurements acquired

from the In-service Aircraft for a Global Observing System database (IAGOS - previously named MOZAIC - Marenco et al., 1998; Petzold et al., 2015). The authors acknowledged that IAGOS offers a relevant reference for characterizing humidity in the UTLS, especially for contrail applications, and used these observations to exhibit main biases on ISSR in NWP models.

Pointwise or neighborhood-based binary metrics have also been applied (e.g. Tompkins et al., 2007; Thompson et al., 2024; Wolf et al., 2025) to evaluate ISSR forecasts discrimination capabilities and compare forecasts against each other. Indeed, it

is both important to discriminate between the capacity to detect ISSR and also to avoid false alarms, due to the trade-off of additional $CO_2$ emissions when avoiding an ISSR. Verification of $RH_{ice}$ and threshold-based conditions is needed to accurately describe NWP model capabilities. ISSR occurrences are known to be an occasional phenomenon in the atmosphere with patchy structures (Spichtinger and Leschner, 2016) embedded in thin layers close to the cold and humid tropopause (Petzold et al., 2020), which requires dedicated metrics including spatialization. For this purpose, the use of available information along

IAGOS aircraft flight paths in the verification process seems valuable and adapted to aviation applications. In addition, recent studies suggested that unreliable forecasts may still have skills to discriminate ISSRs, provided that post-processing calibration or adapted threshold on relative humidity are applied (Teoh et al., 2022; Wolf et al., 2025). This leads to consider a range of different thresholds for ISSR verification for a complete evaluation and comparison of NWP models capabilities. In the end, a shared framework for the verification of ISSR should be defined, which is an important cornerstone of any strategy for avoiding

areas conducive to the triggering of persistent warming contrails.

The proposed study addresses two major questions related to forecasting high-tropospheric humidity and ISSR. The first issue is the development of supersaturation in a global NWP model, the ARPEGE global NWP model of Météo-France, by modifying its cloud scheme. The second aspect addresses the verification of ISSR forecasts using IAGOS humidity observations.

The article is structured as follows. Section 2 is dedicated to the modified Smith cloud scheme (Smith, 1990) used in

ARPEGE for allowance of supersaturation with respect to ice. Section 3 is dedicated to the presentation of observational and forecast datasets used for calibration and verification, and the interpolation methodology. The calibration procedure of the modified cloud scheme is presented in Sect. 4. The verification methodology and associated statistical results are presented and applied in Sect. 5, including a case study. The methodologies and the results presented in this paper are discussed in Sect. 6. Conclusions and perspectives of this work are given in Sect. 7. All the calculations related to the modification of the ARPEGE

cloud scheme are given in the Appendix.

## 2   Modified cloud scheme for the ARPEGE model to allow ice supersaturation

The microphysical scheme used in ARPEGE was first developed by Lopez (2002). It mainly follows the approach proposed in Fowler et al. (1996), but with a reduced complexity. It is based on a prognostic representation of four condensed water species (cloud droplets, ice crystals, rain, snow) for large scale and shallow convective clouds, and on a diagnostic representation

for deep convective clouds. A statistical cloud scheme using a symmetrical triangular probability density function (Smith, 1990) provides cloudiness and mean cloud water condensate within a grid-box for large scale clouds, assuming that cloud condensation and evaporation are instantaneous and reversible. The width of the probability density function is defined by a critical relative humidity threshold that decreases with model level height and horizontal resolution. Partitioning between liquid and ice contents is temperature-dependent, so that the mixed phase lies between 273.15 K and 250 K (more details can

be found in Roehrig et al., 2020). Cloud formation initiates when saturation is locally exceeded within the grid-box, which is

an actual limitation of the ARPEGE cloud scheme as it removes any supersaturation. This hypothesis, validated in the liquid phase, is known to be erroneous in the ice phase, as supersaturation relative to ice is commonly measured by in situ or remote sensing instruments (e.g. Krämer et al., 2009; Heymsfield et al., 2017). Here, we propose a simple extension of the Smith cloud scheme implemented in the ARPEGE model to allow ice supersaturation at cruise levels.

## 2.1 Parametrization of supersaturation w.r.t. ice

The parametrization of supersaturation w.r.t ice in the UTLS is based on the assumption that ice crystals are formed in most cases by the homogeneous nucleation process when the temperature is lower than 235 K. We introduce a saturation ratio coefficient $k$ which is a function of the average temperature in the grid-box $\bar{T}$ and represents the ratio of saturation to be reached to have homogeneous nucleation processes activated in the grid-box. For temperatures colder than 235 K, the saturation ratio coefficient is set such as

$$k = C_{\text{calib}}.f(\bar{T}), \quad f(T) = 2.583 - T/207.8\,K \tag{1}$$

where $f(T)$ represents the homogeneous nucleation threshold as a function of $T$ (see e.g. Koop et al., 2000; Kärcher and Lohmann, 2002; Tompkins et al., 2007; Sperber and Gierens, 2023), and $C_{\text{calib}}$ is a calibration coefficient used for the tuning of the model. From 235 K, a linear return is made back to $k = 1$ at 250 K ($k = 1$ for $\bar{T} \geq 250$ K). This implies that the range of temperature where supersaturation is allowed in our study is set for temperatures colder than 250 K collocating with the pure ice phase limit within ARPEGE microphysical scheme, this range being compatible with persistent contrail applications. In higher temperature conditions where mixed-phase or pure liquid conditions can exist, the ARPEGE cloud scheme is not directly impacted by the modification.

The second step is to write the local thermodynamic adjustment equation to include the saturation ratio coefficient in the classical scheme. The following assumptions are used:

- when ice supersaturation conditions are allowed (i.e. $k>1$), condensate can exist within a grid-box for any point where the saturation ratio $k$ is locally exceeded.

- once the supersaturation threshold is locally exceeded, local adjustment is instantly obtained back to saturation.

Local equations of the cloud scheme are then given by the following set of equations:

$$q_c^+ = \begin{cases} q_t - q_{sat}(T,p) & q_t > k \cdot q_{sat}(T,p), \\ 0 & q_t \leq k \cdot q_{sat}(T,p), \end{cases} \tag{2}$$

where $q_c^+$ is the local condensed content after adjustment, $q_t$ is the total specific humidity before adjustment, that is the sum of the vapor and the condensate ($q_v + q_c$), and $q_{sat}(T,p)$ is the saturation specific humidity w.r.t liquid water or ice before adjustment at temperature $T$ and pressure $p$. The symbol "$+$" denotes that the variable is being diagnosed from the prognostic variables, which means that the value is obtained after adjustment. Figure 1 provides a graphical representation of Eq. (2). With

the modification of the scheme, the total specific humidity (in blue) must exceed the values of $k \cdot q_{sat}$ (in purple) for a cloud to

form, while in the operational version, a cloud would form once $q_{sat}$ (in green) is exceeded. Consequently, areas classified as

cloudy in the operational scheme may be supersaturated with the new scheme.

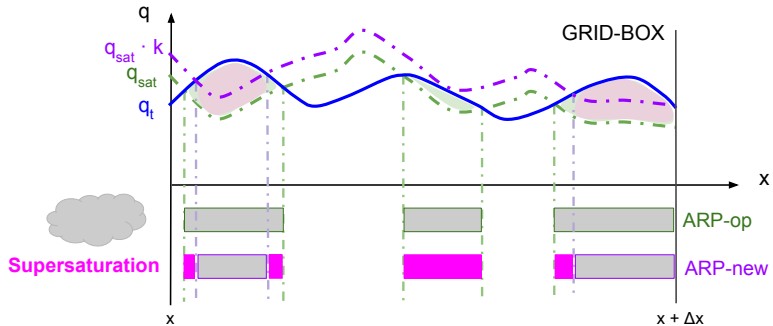

**Figure 1.** Illustration of the variability of $q_t$ and $q_{sat}$ inside a NWP model grid-box. In the current ARPEGE model (denoted ARP-op in
green), thermodynamical adjustment is done back to saturation when $q_t > q_{sat}$. In the modified ARPEGE model (denoted ARP-new in
purple), thermodynamical adjustment is done back to saturation when $q_t > k.q_{sat}$, which allows supersaturation.

## 2.2    Calculation of the mean condensed content and the mean cloud cover

The difference between the modified and the original ARPEGE scheme relies in the introduction of variable $k$ in Eq. (2) to

define the nucleation threshold, and the following consists in reworking the original cloud scheme equations with $k > 1$. To

calculate the mean condensed content and the mean cloud cover in the grid-box, a statistical description of the distance to

supersaturation $q_t - kq_{sat}$ in the grid-box is used. We introduce the notation $\bar{x}$ (resp. $x'$) to represent the grid-box mean value

(resp. the local perturbation) of a variable $x$. Following the cloud distribution concepts developed by Sommeria and Deardorff

(1977) and Mellor (1977), and assuming that pressure fluctuations are negligible in the grid-box (Bougeault, 1981), the local

difference between $q_t$ and $k\,q_{sat}(T,p)$ can be expressed by

$$q_t - k\,q_{sat}(T,p) = Q_{c,k} + s, \tag{3}$$

where $Q_{c,k}$ characterizes the average distance to supersaturation ($k > 1$) resp. saturation ($k = 1$) in the grid-box, such as

$$Q_{c,k} = a_{L,k}\left[\bar{q}_t - k\bar{q}_{sat}(\bar{T}_L,\bar{p})\right], \tag{4}$$

and $s$ represents the local deviation around the value of $Q_{c,k}$ and can be described by

$$s = a_{L,k}\left[q'_t - k\,b_L T'_L\right], \tag{5}$$

with $L$ being the latent heat of condensation, $c_p$ the specific heat capacity of air at constant pressure, and $T_L = T - (L/c_p) \cdot q_c$
the liquid water temperature which is conserved through the condensation process. The factors $a_{L,k}$ and $b_L$ account for changes

in $q_{sat}$ due to latent heating and are given by

$$a_{L,k} = (1 + k\, b_L L/c_p)^{-1}, \tag{6}$$

$$b_L = \varepsilon L \bar{q}_{sat}(\bar{T}_L, \bar{p})/R\bar{T}_L^2, \tag{7}$$

where $R$ is the dry air specific gas constant and $\varepsilon$ is the ratio of dry air constant and vapor constant.

The local deviation $s$ is statistically modeled by a centered $G$-distribution with a standard deviation $\sigma_{s,k}$ such as $s \sim G_{[0,\sigma_{s,k}]}$. This implies that $q_t - k\, q_{sat}$ follows a $G$-distribution with a mean equal to $Q_{c,k}$ and a standard deviation $\sigma_{s,k}$ which gives

$$q_t - k\, q_{sat} \sim G_{[Q_{c,k}, \sigma_{s,k}]}. \tag{8}$$

The mean cloud fraction $C^+$ and mean cloud condensate content $\bar{q}_c^+$ after adjustment can be expressed in relation to the centered and reduced probability distribution $G_{[0,1]}(t)$ (see calculations in Appendix A), such as

$$C^+ = \int\limits_{-Q_{c,k}/\sigma_{s,k}}^{+\infty} G_{[0,1]}(t)\, dt, \tag{9}$$

$$\bar{q}_c^+ = \sigma_{s,1} \int\limits_{-Q_{c,k}/\sigma_{s,k}}^{+\infty} G_{[0,1]}(t)\left(t + \frac{Q_{c,1}}{\sigma_{s,1}}\right) dt. \tag{10}$$

It is important to note that steps from Eq. (2) to Eq. (10) do not depend on the probability distribution that is chosen, hence the methodology can be applied to cloud schemes based on the same type of statistical distribution concepts.

## 2.3 Application to the ARPEGE cloud scheme

The Smith (1990) cloud scheme used in ARPEGE assumes a symmetric triangular probability function for the setting of the $s$ distribution. Applying this methodology in our context, the probability density function is represented by a symmetric triangular distribution with a finite support, its lower and upper limits spanning from $-\sqrt{6}\sigma_{s,k}$ to $\sqrt{6}\sigma_{s,k}$, as represented in Fig. 2. The cloud fraction is non-zero when $-Q_{c,k} < \sqrt{6}\sigma_{s,k}$ indicating the presence of condensed water when the critical point corresponding to $-Q_{c,k} = \sqrt{6}\sigma_{s,k}$ is obtained. This equality can be used to define the critical relative humidity threshold $\bar{U}_{c,k} = \bar{q}_v/\bar{q}_{sat}(\bar{T}, \bar{p})$. Evaluating $Q_{c,k}$ at this critical point where $\bar{q}_t = \bar{q}_v$ and $\bar{T}_L = \bar{T}$ leads to

$$\sigma_{s,k} = \frac{a_{L,k}}{\sqrt{6}} \bar{q}_{sat}(k - \bar{U}_{c,k}). \tag{11}$$

This expression exhibits a direct relation between the width of the distribution $\sigma_{s,k}$ and the critical relative humidity $\bar{U}_{c,k}$. This allows the cloud scheme closure by the definition of a critical relative humidity threshold, as the $s$ distribution is fully determined by its standard deviation. Defining $\alpha_k = Q_{c,k}/\sigma_{s,k}\sqrt{6}$, the integration of Eq. (9) and (10) gives an explicit algorithm (see Algorithm 1) for the computation of $C^+$ and $\bar{q}_c^+$ (see appendix A for calculations). Setting $k \equiv 1$ returns to original Smith cloud scheme.

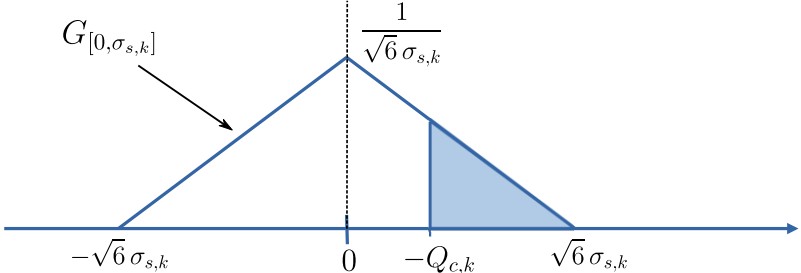

**Figure 2.** Probability density function $G$ applied for the statistical representation of variability inside the grid-box in the modified Smith cloud scheme. Shaded area measures the cloud fraction as it corresponds to the integration of the cloudy part of the grid-box (i.e. $q_t - k\,q_{sat} > 0$).

---

**Algorithm 1** Modified Smith Cloud Scheme

---

If $\quad \alpha_k \leq -1 \quad$ then

$$C^+ = 0 \quad \text{and} \quad \bar{q}_c^+ = 0 \tag{12a}$$

If $\quad -1 < \alpha_k \leq 0 \quad$ then

$$C^+ = \frac{1}{2}(1+\alpha_k)^2 \quad \text{and} \quad \bar{q}_c^+ = \frac{\sigma_{s,1}}{\sqrt{6}}(1+\alpha_k)^3 + \sigma_{s,1}\sqrt{6}\frac{(1+\alpha_k)^2}{2}(\alpha_1 - \alpha_k) \tag{12b}$$

If $\quad 0 < \alpha_k \leq 1 \quad$ then

$$C^+ = 1 - \frac{1}{2}(1-\alpha_k)^2 \quad \text{and} \quad \bar{q}_c^+ = \sigma_{s,1}\alpha_1\sqrt{6} + \frac{\sigma_{s,1}}{\sqrt{6}}(1-\alpha_k)^3 - \sigma_{s,1}\sqrt{6}\frac{(1-\alpha_k)^2}{2}(\alpha_1 - \alpha_k) \tag{12c}$$

If $\quad \alpha_k > 1 \quad$ then

$$C^+ = 1 \quad \text{and} \quad \bar{q}_c^+ = \sigma_{s,1}\alpha_1\sqrt{6} \tag{12d}$$

---

The final closure of the system of equations can be given by an arbitrary definition of $U_{c,k}$ for all values of $k$. However, some simple limiting cases in the $s$ formulation given in Eq. (5) can be exhibited and allows to restrict the closure to the choice of $U_{c,1}$:

**Closure 1** - If predominance of humidity variability term over temperature variability term is assumed in the $s$ formula i.e. $q_t' >> kb_L T_L'$, then $\sigma_{s,k}/\sigma_{s,1}$ is equal to $a_{L,k}/a_{L,1}$ and closure is given by:

$$\bar{U}_{c,k} = \bar{U}_{c,1} + k - 1. \tag{13}$$

**Closure 2** - If temperature variability term is assumed to be predominant in the $s$ formula i.e. $kb_L T_L' >> T_L'$, then $\sigma_{s,k}/\sigma_{s,1}$ equals $k\,a_{L,k}/a_{L,1}$ and closure is given by:

$$\bar{U}_{c,k} = k\,\bar{U}_{c,1}. \tag{14}$$

## 3    Historical dataset

This study aims to compare forecasts issued from several NWP models: ARPEGE with the modified cloud scheme, non-modified ARPEGE, and IFS model from ECMWF, to in situ measurements of UTLS humidity. The observational dataset is issued from instrumental data from the In-service Aircraft for a Global Observing System (IAGOS; https://www.iagos.org). The study period extends from the 1st July 2022 to the 30th June 2023 within the aerial boundary of 80° W–40° E and 30°–75° N, covering the space of North Atlantic and Europe that is one of the world's densest air traffic regions. The vertical domain extends from flight levels FL250 to FL450 (i.e. 375hPa to 150hPa) which corresponds to altitudes 25000ft and 45000ft in the ICAO standard atmosphere (ICAO, 1993), encompassing regions favorable to the triggering of persistent contrails. The datasets are randomly split into two parts, such that half of the flights are used to calibrate the modified ARPEGE cloud scheme, while the other half is used for the verification of NWP model forecasts (see Fig. 3).

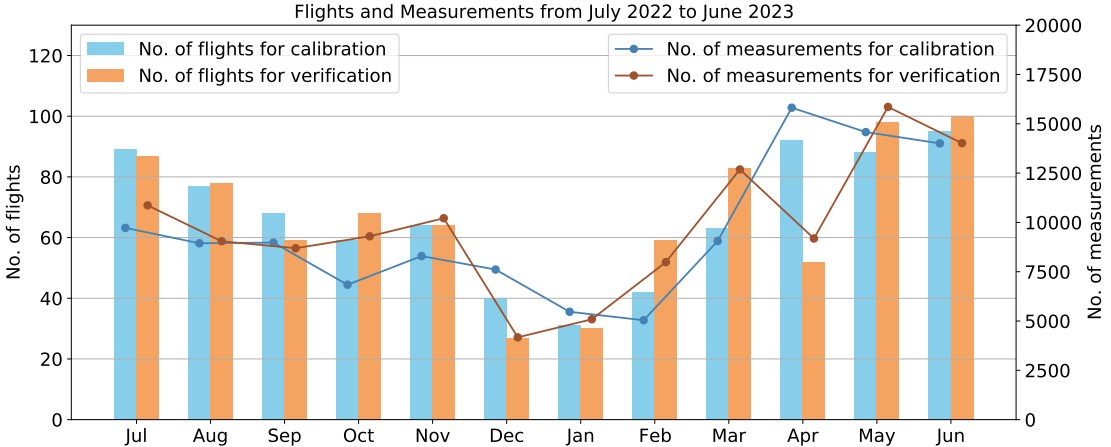

**Figure 3.** IAGOS equipped flights and measurements from July 2022 to June 2023. The dataset is divided in two subsets: flights dedicated to the calibration of the new cloud scheme (blue) and flights dedicated to the verification (orange).

### 3.1    NWP data

The operational version of the ARPEGE model runs 4 times a day with a stretched native grid of 5 km above Europe and 25 km over Oceania and with a 105 level vertical grid. In the following we will denote "ARP-new" the ARPEGE model with the modified cloud scheme, and "ARP-op" the version of the model with the operational (non-modified) cloud scheme. We also perform the verification of $RH_{ice}$ provided by the high resolution (HRES) configuration of the IFS model operated by ECMWF (ECMWF-documentation, 2025, cy48r1). It has a mean native resolution of approximately 9 km worldwide and 137 vertical levels. ARP-op and IFS forecasts start from their own respective operational analysis. The ARP-new forecast starts from the analysis of ARP-op.

For all NWP models, runs of 00Z and 12Z are used, and hourly time steps from T+6 to T+17 are considered for output lead terms. Each output is interpolated on a regular 0.25° lat/lon horizontal grid. The vertical output grid is composed of 21 vertical pressure levels corresponding to flight levels ranging from FL250 to FL450 with intervals of 1000 ft.

## 3.2 IAGOS

IAGOS is a European non-profit association involving research organizations, universities and weather services from Germany, France and the U.K, which provides in situ commercial aircraft observations in relation with air quality and climate change (e.g. Bundke et al., 2015; Filges et al., 2015; Nédélec et al., 2015; Petzold et al., 2015). Ambient air temperature and relative humidity measurements are one of the main IAGOS-core data products. They are measured by a platinum resistance sensor (Pt100) and a capacitive sensor (Humicap-H, Vaisala, Finland), respectively, which are combined into the so-called ICH sensor

(IAGOS Capacitive Hygrometer (Neis et al., 2015)). The accuracy of the instruments used for RH measurements is considered approximately 5 % with typical 1-3 min instrument response time ($\simeq 15 - 50$ km of flight if a cruising speed of 250 m/s is considered) (e.g. Petzold et al., 2020). Aircrafts equipped with IAGOS sensors operate at a cruising altitude between 9 and 13 km, providing a comprehensive in situ dataset of UTLS humidity on a global scale, although northern extra-tropical and, in particular, Europe and Northern Atlantic flight paths are more frequently sampled.

The IAGOS observational dataset and NWP models have very different resolutions. Indeed, the horizontal outputs of the model data is 0.25° (20–25 km at mid-latitudes), while the output frequency of the observations is 4 s along the flight track, resulting in a spatial resolution of approximately 1 km between adjacent observed points, considering regular cruise speeds. In order to have comparable resolutions for the needs of calibration and verification, the IAGOS dataset is smoothed and under-sampled. A temporal interval of 100 s with a centered mean-filter window is applied to obtain an under-sampled resolution of

$\sim$25 km. Nearest point interpolation in time and space dimensions is then used to assign each observation to the NWP outputs in closest proximity.

## 4   Calibration of the modified cloud scheme

In this section, we present the calibration process for the modified cloud scheme. The calibration is based on adjusting the saturation ratio coefficient $k$ through the calibration coefficient $C_{calib}$ (see Fig. 4). The value of $C_{calib}$ is obtained empirically

by comparing the predicted and observed distributions of $RH_{ice}$ using a dedicated calibration dataset. It is also necessary to define a critical humidity threshold for any value of $k$ in order to obtain a closed system of equations. Two formulations given by closure 1 and 2 in equations (13) and (14) are tested (see Sect. 2.3). Both options are set in the basis of the critical humidity threshold $U_{c,1}$ that is used in ARP-op ($U_{c,1} \simeq 50\%$ in the study domain).

The $RH_{ice}$ results given by ARP-new forecasts are compared to IAGOS in situ measurements in the UTLS, for closure 1

(Fig. 5a) and 2 (Fig. 5b). For both closures, the frequency histogram plot with the associated frequency bias on $RH_{ice} > 100$ % events is analyzed as a function of different $C_{calib}$ values: 1.0, 0.9 and 0.8.

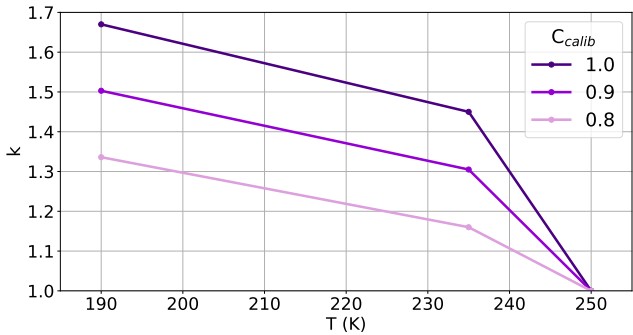

**Figure 4.** Saturation ratio coefficient $k$ as a function of temperature for different values of $C_{calib}$.

The results in Fig. 5 show that supersaturation with respect to ice is achieved with both closures. With closure 1, slightly higher supersaturation values are reached, however, histogram curves show a more pronounced divergence from IAGOS for the different values of $C_{calib}$. In contrast, closure 2 approaches the observations more closely for all RH$_{ice}$ values. We conclude that in our study the closure 2 is better suited to the IAGOS observations and will therefore analyze its results in greater detail.

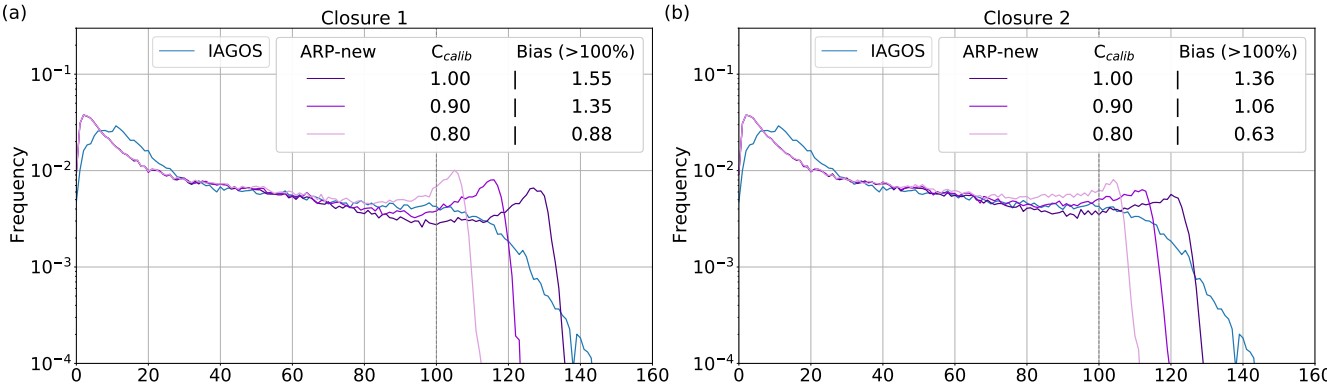

**Figure 5.** Frequency histograms of RH$_{ice}$ (1 % bins) with the associated frequency bias on RH$_{ice}$ > 100 %. Results are shown for under-sampled IAGOS observational dataset and ARP-new for different $C_{calib}$ values ranging from 0.8 to 1.0. In panel (a), ARP-new is defined with closure 1 and in panel (b), with closure 2. Calibration dataset from the 1$^{st}$ July 2022 to the 30$^{th}$ June 2023 within the aerial boundary of 80° W–40° E and 30–75° N, covering North Atlantic and Europe.

In the RH$_{ice}$ histogram shown in Fig. 5b, the ARP-new results demonstrate a high degree of alignment with observations for RH$_{ice}$ values between 80 % to 100 %, especially for $C_{calib} = 0.90$. For supersaturated values ranging from 100 % to 110 %, RH$_{ice}$ values are slightly over-forecast. At a certain point, ARP-new is unable to predict higher supersaturation values. When $C_{calib} = 0.90$ is set, there is a sudden halt around 115 %, meaning that RH$_{ice}$ values above 115 % are not forecast. This limit can be extended to higher values as $C_{calib}$ increases but leads to an over-forecast of ISSR events as shown by the frequency bias results.

The evaluation of the results leads us to select the following calibration set for ARP-new: $C_{calib} = 0.90$ with closure 2. In addition to demonstrating a relative humidity frequency distribution that closely aligns with the observations, this combination has the best bias on RH$_{ice}$ events > 100 % and manages to predict higher supersaturated values than $C_{calib} = 0.80$, without under-predicting, unlike $C_{calib} = 1.00$, the sub-saturated values.

## 5   Verification

This section is dedicated to the verification of RH$_{ice}$ forecast and, more particularly, ISSR events by the calibrated ARP-new model (closure 2 and $C_{calib} = 0.90$). The results are compared to the ARP-op model and the IFS model. We pay particular attention to the use of appropriate and easy-to-interpret metrics in the context of persistent contrail avoidance. Indeed, ISSR occurrences are acknowledged to be an occasional phenomena in the atmosphere (~10 % of the IAGOS dataset in this study) requiring metrics adapted to such imbalanced datasets. Moreover, ISSRs can often have small dimensions compared to the synoptic scale or be organized in heterogeneous patterns (Spichtinger and Leschner, 2016; Gierens and Vázquez-Navarro, 2018), requiring metrics involving a neighborhood approach also referred as "spatial metrics". In addition, the choice of appropriate RH$_{ice}$ decision thresholds for contrail avoidance application is an on-going question (e.g. Dietmüller et al., 2023), demonstrating the need for sensitivity analysis. Then, after a first general analysis of the results on the continuous RH$_{ice}$ variable in Sect. 5.1, we propose the introduction of spatial verification metrics using the available trajectory information in Sect. 5.2, thus allowing a better assessment of ISSR forecast quality in terms of spatial scale. A methodology for a comprehensive evaluation of the discrimination capabilities of ISSR events with different degrees of intensity is also introduced by performing a sensitivity analysis on RH$_{ice}$ thresholds in Sect. 5.3.

### 5.1   Verification of RH$_{ice}$ continuous variable

The RH$_{ice}$ histogram with the associated frequency bias for RH$_{ice}$ > 100 % (Fig. 6a) and the Mean Absolute Error (MAE) (Fig. 6b) for different observed humidity categories are calculated for the verification dataset and for three models: ARP-new, ARP-op and IFS. Model results are compared with filtered and unfiltered IAGOS observations, the latter being used to compute the distribution plots of direct model errors (Fig. 6c).

Figure 6a shows that ARP-new permits supersaturation, exhibiting a close alignment with the observed RH$_{ice}$ histogram, although it does not allow supersaturation exceeding 115 %, as underlined in the calibration study (Sect. 4). The frequency bias of ARP-new for RH$_{ice}$ events greater than 100 % is neutral. ARP-new offers enhanced reliability in comparison to ARP-op, which shows a peak at 100 %, standing out from the rest of the curve and followed by a precipitous drop beyond 100 %, indicating that supersaturation is not forecast. Moreover, ARP-new gets to remove the over-representation of the 70–100 % RH$_{ice}$ categories of ARP-op. Concerning IFS, as shown in previous studies (e.g. Gierens et al., 2020; Sperber and Gierens, 2023), the results show a pronounced peak at 100 % and an under-estimation for higher RH$_{ice}$ values evidenced by the frequency bias (0.81) on RH$_{ice}$ events greater than 100 %. IFS has a significant shortcoming when confronted with values exceeding 110

%. We note that there is no significant difference between filtered and unfiltered observation histograms, showing that applying a 100s mean-filter does not alter the properties represented in the original IAGOS dataset.

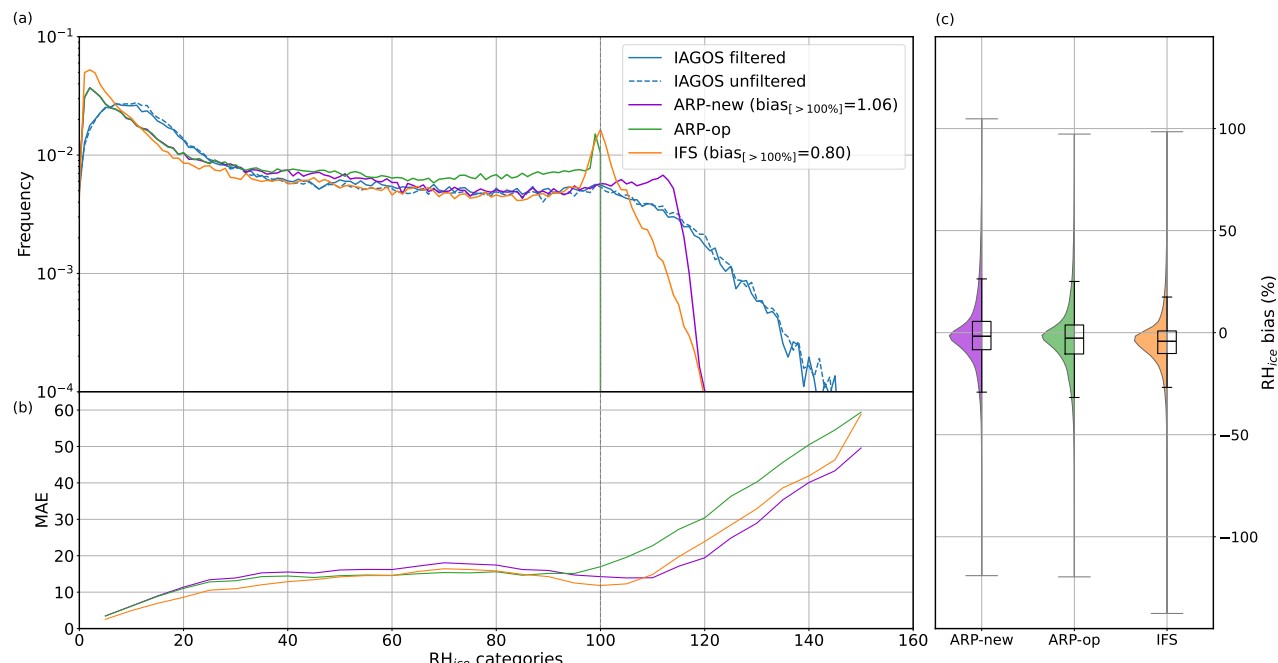

**Figure 6.** (a) Frequency histogram of $RH_{ice}$ (1 % bins) with the associated frequency bias on $RH_{ice} > 100$ % and (b) Mean Absolute Error (MAE) computed for different categories of observed humidity (5 % bins). Results are shown for IAGOS observational dataset (blue), filtered and unfiltered, ARP-new (purple), ARP-op (green) and IFS (orange). (c) Distribution plot of model bias in $RH_{ice}$, computed against unfiltered IAGOS observations, for the three ARP-new, ARP-op and IFS models. Verification dataset from the 1st July 2022 to the 30th June 2023 within the aerial boundary of 80° W–40° E and 30–75° N, covering North Atlantic and Europe.

While the histogram gives a general idea of the distribution of the data, the MAE analyzes each observed $RH_{ice}$ point and its corresponding forecast as to how close they are in value and provides an average of the pointwise model's accuracy. Therefore, the conclusions of both are complementary. In Fig. 6b, ARP-new shows lower MAE values than ARP-op for $RH_{ice}$ categories above 95 %, while in lower categories the MAE is just slightly higher. This suggests that most of the values removed from the over-representation of the 70-100 % categories in ARP-op, have been correctly adjusted to represent supersaturated values in
ARP-new. For categories ranging from 95 to 105 %, the IFS model shows the lowest MAE values across all models, however, exhibiting a steady increase for categories above 105 %. In fact, categories around 100 % are significantly over-forecast by IFS, ensuring a lower amount of error for this specific categories of observations. In return, there is a strong MAE increase for under-predicted categories over 105 %. A similar trend is observed in ARP-new, however, for higher $RH_{ice}$, around 115 %, i.e. precisely when the histogram reveals a shortcoming. As a result, ARP-new exhibits the lowest MAE for higher supersaturated
values.

In the analysis of the direct model error, Fig. 6c shows that ARP-new has a median bias centered on 0, slightly better than the median bias of ARP-op and IFS. However, IFS shows a narrower interquartile range (11.06%), compared to (13.87%) for ARP-new. This indicates that, after bias correction, IFS has a slightly better overall accuracy in representing humidity.

## 5.2 ISSR verification with a trajectory-neighborhood approach

### 5.2.1 Methodology

This section introduces a trajectory-neighborhood verification approach for the evaluation of ISSR and presents resulting scores. In traditional verification methods or point-by-point metrics, perfect scores require near-perfect spatial and temporal matching with observations. However, it is not realistic to expect grid-scale accuracy in well-resolved NWP models due to the presence of small-scale variability (Schwartz, 2017). In fact, even if the model accurately forecasts the size and structure of an existing ISSR, it may be penalized if the predicted ISSR is slightly displaced in space and/or time. This is known as a "double penalty" (Gilleland et al., 2009; Bouttier and Marchal, 2024), as it would be regarded as both a miss and a false alarm, resulting in a less favorable score than if the model had not even predicted the existence of the ISSR. It is therefore useful to implement an effective neighborhood tolerance approach in order to deal with double penalty and capture relevant estimations of the quality of the forecast. From another perspective, neighborhood methods can also be used to identify the spatial scale at which a forecast becomes skillful or to determine the scale at which a desired model skill is achieved.

In this paper, we propose a trajectory-based neighborhood approach that compares predicted values and observations with a neighborhood tolerance oriented upstream and downstream of the flight trajectory. A symmetric approach, illustrated in Fig. 7, is taken to redefine traditional pointwise metrics used for binary event verification such as the hit rate (HR) and the false alarm ratio (FAR) scores (JWGFGR, 2017).

– Hit rate (HR) with neighborhood tolerance: if an ISSR is observed in the flight trajectory, the associated forecast is classified as a true positive if the ISSR is forecast at any point within a distance 'd' upstream or downstream in the flight trajectory. The hit rate in a 'd' neighborhood can then be calculated as the ratio between the number of hits and the total number of ISSR observations. The hit rate is also called *Probability Of Detection* (POD) or *Recall* in literature.

– False alarm ratio (FAR) with neighborhood tolerance: symmetrically, if an ISSR is forecast in the flight trajectory but not observed within a distance 'd' upstream or downstream from the forecast, then it is scored as a false alarm. The FAR is defined as the ratio between the number of false alarms and the total number of ISSR forecasts, and is related to the precision score by Precision $= 1 - \text{FAR}$.

The mathematical expression of HR and FAR with neighborhood tolerance is given by

$$\text{HR} = \frac{\sum_i \max_{k \in \mathcal{N}_{d,i}} F_k \cdot O_i}{\sum_i O_i}, \qquad \text{FAR} = \frac{\sum_i \max_{k \in \mathcal{N}_{d,i}} O_k \cdot F_i}{\sum_i F_i}, \qquad (15)$$

where $F_i$ and $O_i$ are equal to 1 when an ISSR forecast (resp. observation) occurs at $i^{\text{th}}$ record of the trajectory, and where $\mathcal{N}_{d,i}$ represents trajectory-neighborhood records downstream and upstream of the $i^{\text{th}}$ record within a distance $d$ and at constant flight level (i.e. tolerance of $\pm 1000$ft).

Hit rate and FAR are two metrics which characterize the discrimination skills of the forecast system on the positive event in a very intuitive way: the hit rate measures the capability of the forecast system to detect observed ISSRs while FAR represents the predicted ISSR events which actually did not occur. These two complementary metrics often show an inverse relationship, where improving one of them worsens the other and can be combined in a single metric which is commonly used for imbalanced datasets where the emphasis is on the positive event: the $F_\beta$-score (e.g. Christen et al., 2023; Bouttier and Marchal, 2024). This score is given by the weighted harmonic mean of hit rate and precision score, such that

$$F_\beta\text{-score} = \frac{1+\beta^2}{\beta^2\,\text{HR}^{-1} + (1-\text{FAR})^{-1}}, \qquad \left( F_1\text{-score} = \frac{2}{\text{HR}^{-1} + (1-\text{FAR})^{-1}} \right), \tag{16}$$

where $\beta > 1$ gives more weight to detection compared to precision and inversely if $\beta < 1$. In the case of rare events with high impact, the emphasis is often on the hit rate. However, in the case of ISSRs, it is important to ensure precision. By avoiding areas that are not actually supersaturated, aviation contribution to global warming could increase with respect to the baseline trajectory by supplementary $CO_2$ emissions, thus achieving the opposite effect to what was intended. In this paper, a neutral value of $\beta = 1$ is chosen and we leave to future studies the possibility of considering other values of this parameter in conjunction with optimization work involving climate metrics.

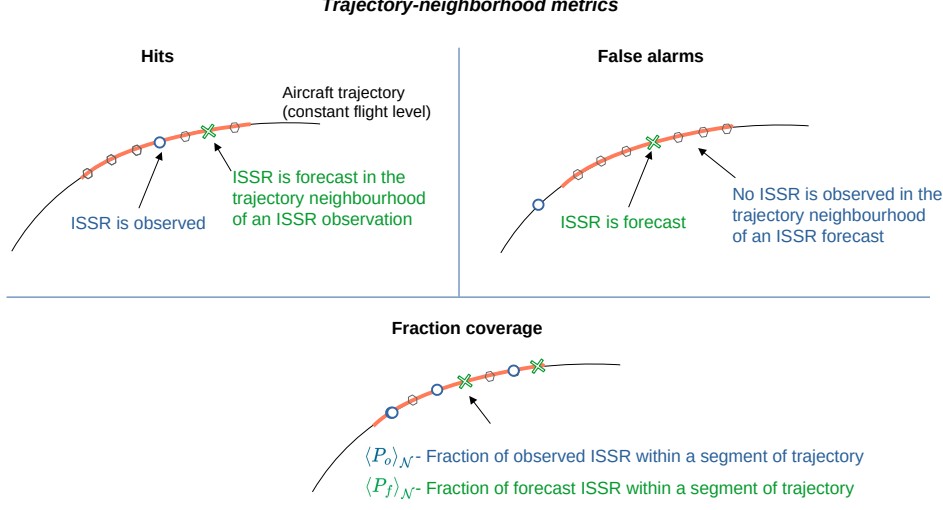

**Figure 7.** Definition of hits, false alarms and fraction coverage methodology on ISSR events with a trajectory-neighbourhood approach.

In addition to the $F_\beta$ score, a trajectory-neighbourhood Fraction Skill Score (FSS) is implemented. This widely used method compares the forecast and observed ISSR fractional coverage within a defined spatial neighborhood (Ebert, 2009; JWGFGR,

2017). It is defined in our context by

$$\text{FSS} = 1 - \frac{\sum\limits_{i}\left(\langle P_f\rangle_{\mathcal{N}_{d,i}} - \langle P_o\rangle_{\mathcal{N}_{d,i}}\right)^2}{\sum\limits_{i}\langle P_f\rangle^2_{\mathcal{N}_{d,i}} + \sum\limits_{i}\langle P_o\rangle^2_{\mathcal{N}_{d,i}}}, \tag{17}$$

where $\langle P_f\rangle_{\mathcal{N}_{d,i}}$ and $\langle P_o\rangle_{\mathcal{N}_{d,i}}$ are the fraction of ISSR forecasts (resp. observations) in the neighborhood $\mathcal{N}_{d,i}$ (see illustration in Fig. 7). The score is a skill metric inspired by the Brier score, which evaluates the difference between observed and predicted ISSR event fractions, considered as probabilities.

### 5.2.2 Results

The results of ISSR verification with trajectory-based approach are given in Fig. 8, where the outcomes of HR, FAR, F$_1$-score
and FSS are shown for varying neighborhood values ranging from the nearest to 240 km, with 30 km increments. The 95 % confidence intervals are calculated by using the bootstrap technique to account for sampling uncertainty (e.g. Bradley et al., 2008).

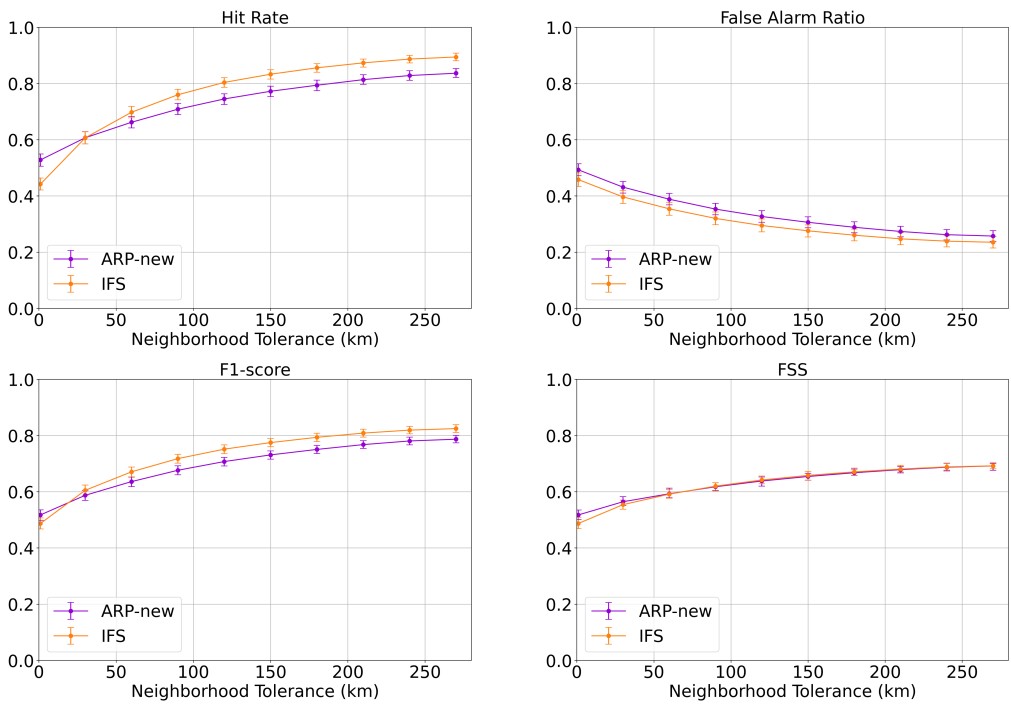

**Figure 8.** HR, FAR, F$_1$-score and FSS score on ISSR events (RH$_{\text{ice}}$ > 100 %) forecasts depending on increasing neighborhood spatial tolerance, for ARP-new (purple) and for IFS (orange) with 95 % confidence intervals. Neighborhood distances range from the nearest to 270 km, with 30 km increments. Verification dataset from the 1st July 2022 to the 30th June 2023 within the aerial boundary of 80° W–40° E and 30–75° N, covering North Atlantic and Europe.

The results show that the scores have a high sensitivity to the neighborhood tolerance. With a neighborhood tolerance of 150 km which represents approximately 10 minutes of flight, ARP-new has a HR of around 77 % and a FAR of 31 %, compared

with 53 % and 49 % respectively for the nearest grid-point. These values indicate that when an ISSR was observed, an ISSR was forecast in a 150 km neighborhood in 77 % of the cases, and symmetrically that when an ISSR was forecast, an event could be observed in 69 % of the cases. If higher neighborhood tolerances are applied, the HR can reach values higher than 80 % with a FAR lower than 30 %, showing a rather good spatial agreement between observations and forecasts of ISSRs events.

A comparison of the discrimination capabilities of ARP-new and IFS shows that both NWP models have close skills. When

no neighborhood tolerance is taken into account, ARP-new has a higher HR but also a higher FAR than IFS. This translates into close $F_1$ and FSS scores for both models, with a slight advantage for ARP-new. Increasing the neighborhood tolerance improves all scores for both models, but IFS shows a better HR, FAR and $F_1$-score than ARP-new, while the FSS scores still overlap. Some care has to be taken when comparing HR, FAR and $F_1$-score with neighborhoods. Indeed, because of its general underestimation of ISSR (frequency bias $\sim 0.8$), IFS will have a tendency to have low FARs and low HRs when the nearest

point is considered. When the neighborhood tolerance is increased, the HR becomes less sensitive to this dry bias, since a single positive forecast in the neighborhood is enough to obtain a hit. This explains why, when the neighborhood tolerance is increased, IFS shows a greater increase in the HR, and consequently, in the $F_1$-score compared to ARP-new which has a neutral bias. Although the FSS score is less easy to interpret, it appears to be more robust for global performance comparison purposes as it does not artificially offset biases.

## 5.3   Sensitivity analysis on $RH_{ice}$ decision threshold

In Sect. 5.2, the $RH_{ice}$ threshold has been set at 100 % in order to identify ISSRs in both forecasts and observations. However, there is a high interest in studying the sensitivity of scores on ISSR threshold definition for both forecast and observations:

- Regarding forecasts, varying the $RH_{ice}$ threshold provides an overview of the NWP model performance range and should be considered for a complete evaluation of NWP model discriminatory skills. Since models are not perfectly calibrated

to observations, there may be forecast threshold values that offer better scores than the 100 % threshold regarding the end-user needs (see e.g. Dietmüller et al. (2023) who used a threshold lower than 100 % for ISSR detection in ERA5 models, and Teoh et al. (2022) and Wolf et al. (2025) who used post-processing recalibration of the $RH_{ice}$ parameter).

- Regarding observations, evaluating the forecast of highly ice supersaturated regions (i.e. $RH_{ice} \simeq 110$ %, 120 %, ...) is also needed, as literature has shown that the lifetime persistence of contrails is influenced by the level of the

supersaturation in the ISSR (Schumann et al., 2012). For slightly subsaturated regions (i.e. $RH_{ice} \simeq 90$ %, 95 %), there is also an interest as it has been shown that saturation rates just below ice saturation could be sufficient to allow the existence of persistent contrails (Li et al., 2023).

In order to perform this sensitivity study, a robust approach in the context of imbalanced datasets is to use a graphical tool called the Precision-Recall curve (Saito and Rehmsmeier, 2015). The Precision-Recall curve is drawn by calculating the recall

(i.e. hit rate) and precision (i.e. 1-FAR) scores at several $RH_{ice}$ thresholds for the forecast. In general terms, the curve closest

to the top/right corner corresponds to the best discrimination capabilities, where precision and hit rate (recall) are optimal, i.e., equal to 1. This graphical tool is roughly equivalent to the widely used Receive Operation Characteristic (ROC) curve, but the Precision-Recall curve is better used in the context of rare events (Saito and Rehmsmeier, 2015; Bouttier and Marchal, 2024).

In Fig. 9, we show the Precision-Recall curves computed with no neighborhood tolerance. We represent four distinct plots, each corresponding to a different $RH_{ice}$ threshold for the observations: 95 %, 100 %, 105 % and 115 %. We include ARP-op in this plot because, although it does not permit supersaturation, it may be capable of detecting ISSRs with a threshold of less than 100 %. For each curve, we calculate the Average Precision (AP), which is defined as the area under the Precision-Recall curve. The AP offers a concise overview of the Precision-Recall curve, facilitating direct comparisons between models and their scores across varying $RH_{ice}$ ranges. Moreover, we represent the diagonal line where neutral frequency bias is obtained.

The results obtained with the observation threshold of 100 % (Fig. 9b) indicate that all three models have close APs, with an advantage for IFS. In addition, IFS exhibits the highest precision and recall scores close to the diagonal where neutral frequency bias is obtained. A comparison of the APs obtained with the 95 % threshold (Fig. 9a) reveals an improvement for all models with IFS being significantly better in the diagonal region. At higher humidity ranges, with the threshold of 105 % and 115 % (Fig. 9c and d), the APs are significantly lower for all the models, with ARP-new and ARP-op now performing the best. When comparing APs obtained for relative humidity above 100 % versus 115 %, a decrease of approximately 60 % in scores is revealed for all models.

The comparison of ARP-new model with ARP-op model shows that both models have a similar ability to discriminate ISSRs. For example, forecast threshold of 90 % in the non-modified ARPEGE corresponds to broadly the same HR and precision values as a threshold of 100 % with the modified cloud scheme. We conclude that the modified scheme strongly enhanced the reliability of the ARPEGE forecast of ISSR (as shown by Fig. 6), while being neutral on its inherent discrimination skills.

A comparison of the monotony of the Precision-Recall curves reveals that ARP-new and ARP-op exhibit divergent behavior compared to IFS. The ARPEGE curves are quasi monotonic, indicating that an increase in the decision threshold of the forecast is accompanied by an increase in precision. In contrast, IFS does not follow the same behavior. From a certain threshold close to 98 %, an increase in the decision threshold results in a decline in both precision and recall. This penalizes the global AP score for IFS, while it exhibits its maximum potential of ISSR discrimination for thresholds close to 98 %.

In addition to providing an overview of the performance of each model, the Precision-Recall curves can also be used to adjust the decision threshold in line with specific preferences regarding non-detections and false alarms. For example, should a pointwise HR of 0.6 be requested, we can ascertain the $RH_{ice}$ threshold that should be selected in each model to discriminate ISSRs and the pointwise precision that will be achieved. If we fix the observation threshold at 100 % and aim for a HR of 0.6, we will have to choose $RH_{ice} = 97$ % as the threshold for the ARP-new forecast, giving a precision of 0.49. Similarly, we would have to choose a threshold of 87 % for ARP-op, with a precision of 0.47, and 97 % for IFS, with 0.55 precision.

Finally, it should be noted that the Precision-Recall curve can also be calculated with a neighborhood tolerance, following the methodology presented in Sect. 5.2.1 to calculate the HR and FAR.

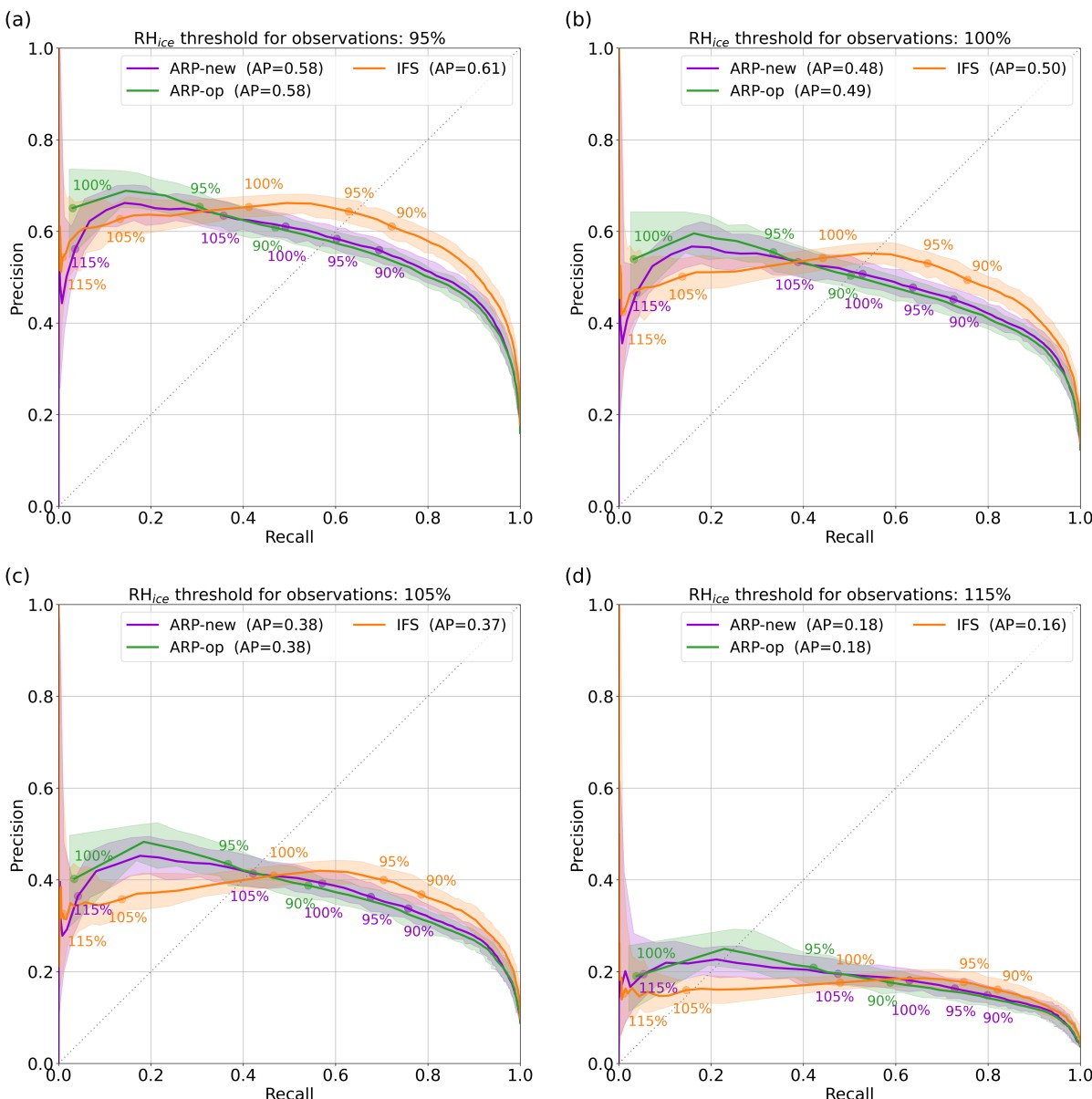

**Figure 9.** Precision-Recall curves representing discrimination skills of ARP-new (purple), ARP-op (green) and IFS (orange) for different RH$_{ice}$ thresholds for observations: (a) 95 %, (b) 100 %, (c) 105 % and (d) 115 %. All the curves have been calculated without neighborhood tolerance. Intervals of 95 % confidence are shown with shaded areas. The points on the curves corresponding to the forecast thresholds are indicated accordingly: 95 %, 100 %, 105 % and 115 %. The legend includes the Average Precision score (AP) for each curve. The diagonal represents the neutral bias line. Verification dataset from the 1st July 2022 to the 30th June 2023 within the aerial boundary of 80° W–40° E and 30–75° N, covering North Atlantic and Europe.

## 5.4 Case study

In this section, we investigate a transatlantic IAGOS flight from 23 September 2022 between Frankfurt (Germany) and Washington D.C. (USA). A comparison is made between the observed and forecast ISSRs using IAGOS measurements and NWP data from both ARP-new, ARP-op and IFS. Figure 10 shows RH$_{ice}$ along the IAGOS flight path. Take-off and landing are not included in this graph, as flight levels below FL250 are not of interest for this study.

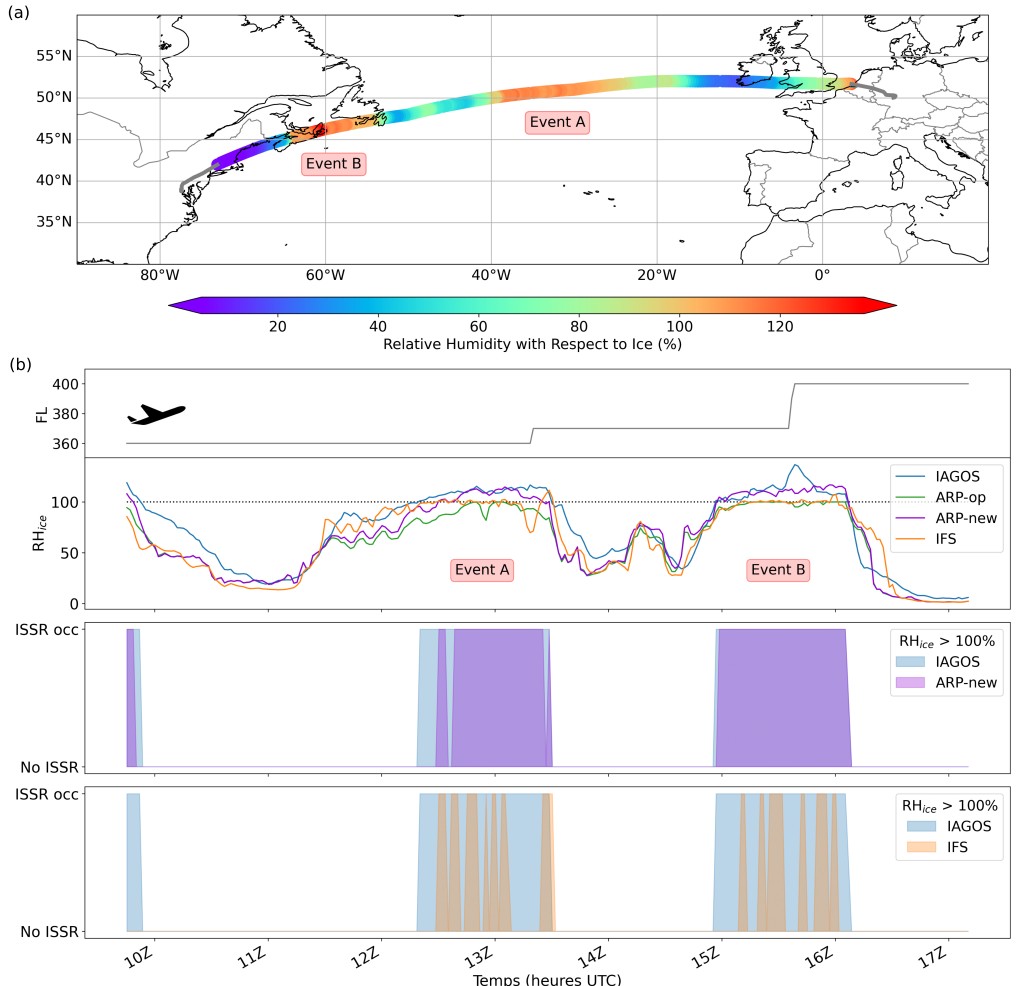

**Figure 10.** Case study of a transatlantic IAGOS flight, departing Frankfurt (FRA), Germany, on 23 September 2022 at 09:07 UTC and landing in Washington (IAD), USA, at 18:03 UTC. Panel (a) presents the flight trajectory with the observed RH$_{ice}$ values. Two main ISSR events are identified denoted $A$ and $B$. In panel (b), the times series shows the temporal evolution of RH$_{ice}$ during the flight trajectory as measured by the IAGOS sensor (blue) and as forecast by NWP models - run 00Z. We represent ARP-new (purple), ARP-op (green) and IFS (orange). The three bottom series highlight observed ISSR occurrences (blue shading) with predicted ones for ARP-new (purple shading) and IFS (orange shading).

There are two distinct areas where the values are greater than 100 %, shown in orange and red (Fig. 10a). The first is over the Atlantic Ocean (region $A$), and the second is the vicinity of Newfoundland, Canada (region $B$). In Fig. 10b, the time

series show the temporal evolution of $RH_{ice}$, with ISSR events highlighted in bottom plots to facilitate comparison between observations and forecasts. In this case, the ARP-new model has a very good agreement with the occurrence of ISSR events in regions $A$ and $B$, particularly in region $B$. In the region $A$, $RH_{ice}$ reveals that the ARP-new curve is slightly below the ISSR threshold while the observations start being supersaturated, yet leading to 'miss' events when, in fact, the global view of the episode is well predicted. This demonstrates the value of the neighborhood approach as a means of correctly interpreting

scores. ARP-op, as it does not predict $RH_{ice}$ values higher than 100 %, does not detect the ISSRs. However, it follows closely the time evolution of observations, even though it remains below the observations for $RH_{ice} > 100$ %. With regard to IFS, while the plot comparing the ISSRs indicates that it did not fully predict the episode, the relative humidity evolution reveals a pattern very closely aligned with the observations, but with a deficiency in reaching higher supersaturated values, confirming the IFS under-prediction of high values of $RH_{ice}$, as shown in Fig. 6. In the case of IFS, introducing a neighborhood tolerance

in the verification process allows to counterbalance this under-representation of ISSR and exhibit the potential of detection capabilities of this model.

Figure 11 provides a zoomed spatial view of regions $A$ and $B$. ISSR observed events in the flight trajectory are compared to ARP-new forecast at corresponding flight levels, and the presence of multiple persistent contrails is shown on the Cirrus Reflectance Product from Suomi NPP/VIIRS at times close to the flight path. This example highlights the good agreement

between observations and forecast and confirms the potential of what could be made using NWP model forecasts to predict environments favorable to persistent contrail triggering.

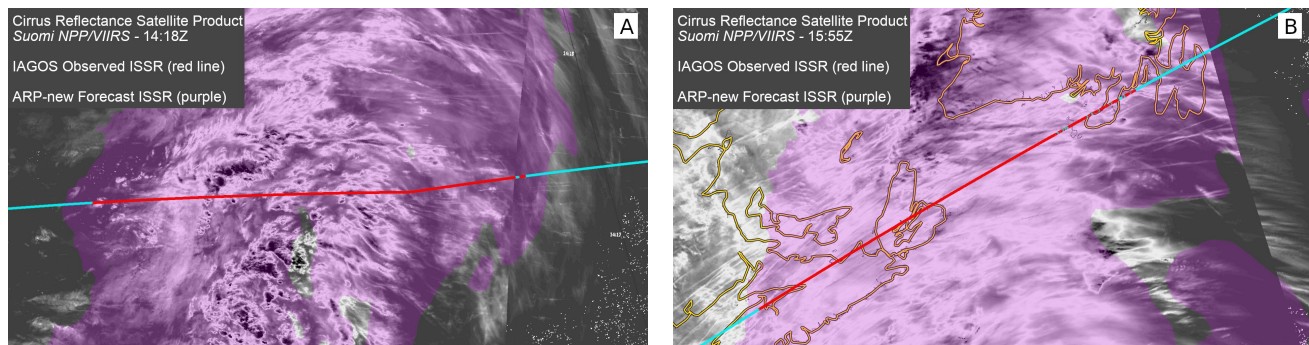

**Figure 11.** Cirrus Reflectance satellite imagery provided by NASA Worldview (https://worldview.earthdata.nasa.gov) from Suomi NPP/VIIRS identifying persistent contrails regions $A$ (left) and $B$ (right) on 23 September 2022 from the case study shown in Fig.10. IAGOS flight path is shown (cyan dots) exhibiting observed ISSR events (red dots). Forecast of ISSR by ARP-new model run-00Z at lead terms T+13 (left) T+16 (right) is given in spatio-temporal vicinity of the flight trajectory.

## 6 Discussion

Discussion on three key areas is proposed in the following. First, we analyze the modifications and assumptions we have made on the cloud scheme and microphysics of the ARPEGE model. Second, we explore the definition of appropriate verification methodologies to provide a comprehensive picture of a model's skill in the context of ISSR avoidance. Finally, we discuss the results obtained on the performance of NWP models for ISSR discrimination and the implications for contrail avoidance applications.

### 6.1 Cloud scheme and microphysics

A modified cloud scheme for ARPEGE NWP global model is presented in this work to enhance the representation of relative humidity w.r.t ice, and in particular supersaturation, which is a necessary condition for contrail persistence. The modeling is based on a generalization of the Smith cloud scheme currently used in the operational ARPEGE. This modification can be implemented without any major modification and does not imply supplementary computational effort. A notable point is that the methodology developed in this article reworks the Sommeria and Deardorff (1977) statistical concepts while incorporating ISSR parametrization, which allows for extensions to other atmospheric models using a similar framework. This is the case, for example, of the ICE3 microphysical scheme implemented in the French regional NWP model AROME (Seity et al., 2011, 2012), which uses a turbulence-based Gaussian distribution for the representation of clouds, and which could benefit from these developments.

Some common assumptions are shared with the parameterization implemented in the IFS model (Tompkins et al., 2007), in particular the use of a temperature function based on Koop et al. (2000) to describe the ice nucleation threshold, and the assumption that once this threshold is locally exceeded, local adjustment is instantly obtained back to saturation in the cloud. However, there are major differences between the two models in terms of their respective microphysical frameworks. For example, cloud fraction is diagnostic in ARPEGE and prognostic in IFS. These different frameworks also implies differences in the representation of subgrid variability: IFS uses a dedicated distribution combining uniform (out-cloud) and Dirac (in-cloud) distributions, compatible with its prognostic cloud fraction parametrization, while ARPEGE uses a triangular distribution. The ARPEGE model also differs from other GCMs that allow supersaturation, such as ECHAM or CAM (see e.g. Bock and Burkhardt, 2016; Neale et al., 2010; Chen et al., 2012). These models are used for climate prediction and include supersaturation w.r.t ice, generally associated with a parametrization of contrail cirrus. In both cases, the triggering of homogeneous nucleation is proposed in a similar manner, mainly driven by a temperature function, but the thermodynamic in-cloud adjustment to saturation is relaxed, within a more complex framework of a 2-moment microphysical scheme.

In the new scheme, some assumptions are made regarding the introduction of supersaturation in ARPEGE cloud scheme, most of them having already been discussed in Tompkins et al. (2007); Sperber and Gierens (2023). For example, in the modified scheme, the adjustment in the cloud to ice saturation is assumed to be instantaneous, which is probably a major limitation of the physical description of ice representation in our context and an explanation of the cut-off on the highest supersaturation observed in the $RH_{ice}$ distribution histogram (Fig. 6a). Indeed, it has been shown that $RH_{ice}$ decreases with a

relaxation time that can exceed several time steps, to finally reach a few percent above saturation, thus allowing local in-cloud supersaturation w.r.t ice. Taking into account pre-existing ice and a better description of the physics of the vapor deposition should then improve the representation of the highest values of ISSR. Sperber and Gierens (2023) proposed a cooling-cloud formation-phase relaxation process as an alternative to instantly adjusting to the equilibrium value. This modification requires the addition of new prognostic variables describing the history of in-cloud supersaturation, which adds further complexity, but could result in a better description of $RH_{ice}$ distribution for the highest values. Regarding cirrus cloud formation, homogeneous nucleation is considered dominant compared to heterogeneous nucleation at the altitudes and temperatures of interest for the contrail application. It can be justified by the presence of relatively pristine environments with few ice nucleating particles (see e.g. Gierens, 2003; Tompkins et al., 2007; Sperber and Gierens, 2023). Thus, a threshold with temperature dependence issued from Koop et al. (2000) is used to represent the maximum local value of supersaturation which can be obtained before homogeneous nucleation processes are activated. Sperber and Gierens (2023) highlight that for slow updrafts the amount of generated ice crystals could however become large enough to reduce the supersaturation limit. This could lead to incorporate the vertical velocity in the expression of this coefficient. Regarding the use of microphysical 2-moment schemes, we acknowledge that cloud formation processes can be better represented in NWP models involving more detailed physical processes (Vié et al., 2016; Thompson et al., 2024; Seifert, 2024). For GCM climate models where contrail clouds are parametrized (e.g. Bock and Burkhardt, 2016; Chen et al., 2012), this type of scheme has proven useful for estimating the physical and optical properties of contrails, such as optical length (see e.g. Zhang et al., 2024). However, there is currently no consensus that obtaining a satisfactory representation of $RH_{ice}$ in the UTLS necessarily requires the introduction of a second moment into the microphysical scheme, and the balance between computational costs and benefits in terms of improved forecasts must be carefully evaluated before implementing such a complex scheme in an operational NWP system.

We decided to focus this study on the Europe and North Atlantic domain due to the high density of aircraft in this area, which provides a large number of available observations (Teoh et al., 2024). However, we believe that this work could be extended to tropical regions. In this case, it could be calibrated accordingly, for example by modifying the $C_{calib}$ coefficient (see calibration procedure, section 4), given that tropical climatology differs from mid-latitude regions and that the occurrence of ISSR depends on factors such as dominant deep convection (Spichtinger and Leschner, 2016). Similarly, North America and Asia are large continental areas with large mountain ranges, and the distribution of processes leading to ISSR in the atmosphere (e.g. warm conveyor belt, convection) is likely to differ from European/North Atlantic climatology, which could alter the results.

The final steps towards operationalizing the modified scheme are to verify the impact on the general parameters of the global NWP model (temperature, wind, etc.) and to adjust all the parameterizations accordingly. Indeed, the introduction of supersaturation implies a reduction of the cloud cover and cloud ice contents in the altitudes which are impacted by the modification. This change will have a direct impact on temperature through changes in radiative and latent heat transfers, with an expected cooling effect, mainly in the upper troposphere (see Appendix B). The intensity of these impacts, their consequences for the dynamics, and whether they benefit or not for ARPEGE will be assessed in a dedicated forthcoming study.

## 6.2 Verification methodology

In this work, particular attention has been paid to the definition of appropriate verification metrics. In addition to traditional verification methods on humidity variable (e.g. Frequency histograms, Mean Absolute Error), the statistical results emphasize the importance of adopting a neighborhood approach when evaluating the capacity of a model to discriminate ISSRs. Introducing a neighborhood tolerance allows to link the quality of the forecast to a spatial scale, as well as addressing several scoring issues (e.g. double penalty, observation uncertainty, interpolation on gridded data). The trajectory-neighborhood method is intuitive, simple to implement and results can be interpreted in the frame of a flight trajectory. However, it does not include forecast information on the adjacent vertical levels, nor of points to the right or left of the flight trajectory. Consequently, the method compromises between maintaining symmetry in dealing with forecasts and observations and ensuring straightforward interpretation with the flight trajectory, while disregarding the potential to consider forecasts of ISSRs that may be offset to one side of the flight trajectory.

The verification metrics HR, FAR and $F_\beta$-score with the use of a neighborhood tolerance, give easily interpretable information on spatial scales at which a certain degree of correspondence between observed and forecast events is achieved, but this methodology can hide biases and should be completed by its estimation to have a complete overview of the performances of the forecasts. The FSS score is less directly interpretable, but provides a summary of all the factors that can assess NWP model quality (including true negatives in the verification). The Recall-Precision plot has shown to be a valuable tool to measure the sensitivity of different NWP models to $RH_{ice}$ decision threshold, and provide a full NWP model inter-comparison outcome. Further verification metrics could also be explored by methods based on neighborhood-populating contingency tables (e.g. Gilleland et al., 2009; Schwartz, 2017; Stein and Stoop, 2019), each method having their qualities and drawbacks regarding for example inconsistent event definitions or keeping symmetry between hit and false alarm calculations. Other advanced spatial verification techniques, such as object-based methods (Wolff et al., 2014) could provide further information on NWP model quality, for example by targeting most significant ISSR features. However, implementing this kind of methodology can present some challenges regarding mathematical data processing to obtain geometrically coherent ISSR features.

In future works, other observational dataset could be useful for further verification of the NWP models. IAGOS dataset is open-accessible and quality-checked, relevant for humidity measurements in the UTLS. However, there is an heterogeneity in sampling due to the operational flight paths of participating airlines (Wolf et al., 2025). The highest measurement density is found across the North Atlantic flight tracks, North-Eastern America and Europe at cruise altitudes (200hPa to 250hPa). In addition, the sampling is biased by avoiding aviation hazards (e.g. deep convective clouds and the outflow of such clouds, turbulent regions) and by route optimization according to the jet stream. Radiosonde observations which mainly sample the atmosphere above continental surfaces represent a valuable complementary database in world regions or weather situations where IAGOS data are less represented. This observation source also makes it possible to assess humidity at levels above or below typical cruising altitudes (see e.g. Bland et al., 2021; Thompson et al., 2024). Also, contrails tracked using satellite imagery, ground-based cameras and lidars (Vazquez-Navarro et al., 2010; Kulik, 2019; Chevallier et al., 2023), along with flight and ground-based contrail observations (Curat and Péchaud, 2023, COOP Program) represent an additional data source.

However, verification is less direct, as both contrail formation (Schmidt-Appleman) and its persistence (ISSR) must be considered together.

## 6.3 Performances of NWP models for ISSR discrimination and potential use for contrail avoidance application

This study shows that ARP-new has an enhanced agreement with $RH_{ice}$ observations from IAGOS measurements in the UTLS when compared to ARP-op. Results of Sect. 5.1 show that it corrects the dry bias of $RH_{ice}$ above 70 % and overcomes the limit at 100 % that the operational model exhibits. However, it over-predicts slightly occurrences of $RH_{ice}$ in the range of 105 % to 115 % and under-predicts values above 115 %. As a confirmation of previous studies, IFS exhibits also a capability to produce supersaturation w.r.t ice, but with an overestimation of values very close to 100 %, a general underestimation of ISSRs, and a shortcoming in representing values above 105 %.

In terms of model performance to detect ISSR events, ARP-new and IFS models show close skills as shown by the FSS score verification in Sect. 5.2.2. For ARP-new, HR and FAR results indicate that when ISSR was observed in IAGOS, an ISSR was forecast in a 150 km upstream or downstream the trajectory (i.e. 10 min of flight) in ∼80 % of the cases. Symmetrically when an ISSR was forecast, ISSR event could be observed in ∼70 % of the cases. This indicates a rather good spatial correspondence between forecasts and observations. For IFS, the scores are even better but must be tempered because of the IFS dry bias which is artificially corrected when neighborhood tolerance is applied. These statistical results are well illustrated by the case study presented in Sect. 5.4. On this case, the correspondence between forecast and observation patterns is exhibited for both ARPEGE and IFS models on $RH_{ice}$ spatio-temporal evolution. On the one hand, ARP-new corresponds closely to observations in the ISSR zones. On the other hand, ARP-op and, to a lesser extent, IFS show underestimation of the highest $RH_{ice}$ values in accordance with their statistical dry bias on ISSR.

The sensitivity study driven on the $RH_{ice}$ threshold provides additional information. In particular, it allows a comparison on NWP model skills for detecting ISSR events, relaxing the constraint on the choice of $RH_{ice}$ decision thresholds in the forecast. This comparison shows that ARP-new outperforms IFS in predicting the highest supersaturated events and that, conversely, IFS outperforms ARP-new when slight undersaturation thresholds are considered. In addition, the range of HR and FAR pairs that can be obtained by each model with adjusted thresholds is measured. It shows that even without allowing supersaturation, the ARP-op has also skill for discriminating ISSRs, similar to ARP-new if decision thresholds lower than 100 % are applied, which is a major conclusion of this work. This suggests that post-processing recalibration of $RH_{ice}$ outputs can be a workable strategy to use non-allowing supersaturation NWP forecast such as ARP-op. Finally, the study shows that IFS and ARP-new recall-precision curves have a shortcoming on precision when decision thresholds greater than a certain amount (98 % for IFS, resp. 110 % for ARP-new) are applied in the forecast. Indeed, while higher thresholds should lead to greater success ratio of ISSR forecasts, the results show a stagnation of precision beyond these thresholds. This is particularly true for IFS leading to the conclusion that applying thresholds greater than 98 % to detect ISSR is de facto sub-optimal. We believe that this behavior should be further analyzed to understand its cause, probably related to the saturation adjustment process.

It is important to note that the comparison between different NWP models which is conducted in this study is not able to separate the specific contribution of the cloud and microphysical schemes in ARPEGE or IFS regarding ISSR forecast. In a

schematic view, the quality of a NWP model for predicting supersaturation w.r.t ice is the result of the quality in representing several processes across various scales, among them: at the largest scales, the global circulation, which induces the positioning of high water content areas in the right geographical location, and at a lower scale the microphysical processes that allows to estimate the right amount of supersaturation reached in these areas. Comparing separately cloud schemes within the same NWP model could be done in dedicated research work but will represent a certain amount of implementation and calibration work. A first, lighter approach, which could be used to minimize the impact of global circulation on scores, would be to base forecasts on the same analysis at initial time.

Moving forward to end-user verification metrics is sketched out in this work by proposing the use of spatial metrics in the verification to take into account the spatial scales of the operational context. This may involve the inclusion of a margin of uncertainty when planning avoidance strategies. For instance, according to Fig. 8, including a 150 km margin to avoid an ISSR forecast helps guarantee that ~80 % of the ISSRs that actually occurred will be avoided, and that ~70 % of the flight paths will be appropriately rerouted, otherwise an ISSR would have been crossed. The introduction of the $F_\beta$-score is also a user-oriented approach as it allows to give differential weight in scoring to detection or precision capabilities of the forecast. Indeed, precision has a great importance in a contrail avoidance operation where supplementary $CO_2$ emissions are emitted.

Assessment of quality regarding contrail application requires a further step beyond this study, involving traffic and use of climate metrics with associated $CO_2$ and non-$CO_2$ contributions. Some additional degrees of freedom in the use of NWP models in the operational context can also be obtained which opens optimization possibilities. For example, the sensitivity study on $RH_{ice}$ decision threshold presented in this work can be used as a base for future optimization work which would tune this threshold according to climate impact metrics and operational constraints. In the same vein, the use of a multi-model approach or/and ensemble forecasts could also prove very useful for this type of trade-off problem by tuning a probability threshold on persistent contrails occurrence conditions.

# 7   Conclusions

This article presents improvements in the cloud scheme of the global ARPEGE NWP model to allow the representation of supersaturation w.r.t ice in the UTLS. This study goes hand to hand with a verification methodology that explores NWP model skills in forecasting $RH_{ice}$, within the context of persistent contrail mitigation applications. Forecasts of ARPEGE with the modified cloud scheme, are compared to the non-modified (operational) ARPEGE and the IFS model from ECMWF which already implements a supersaturation parametrization. The models are evaluated against aircraft in situ measurements of UTLS humidity from the IAGOS program. Here are the main conclusions of the study:

- The modified cloud scheme allows supersaturation and bias correction for humidity, showing close alignment with the $RH_{ice}$ distribution of in situ aircraft measurements. This modification does not require any major algorithmic changes or additional computational effort. In addition, the methodology used provides a framework that can be applied to similar statistical cloud schemes.

- Spatial verification allows to show that a good spatial agreement between forecast and observations is obtained. For example, discrimination capabilities when forecasting ISSRs with the modified ARPEGE at lead terms between 6 and 18 hours show a hit rate of $\sim 80\%$ and a false alarm ratio of $\sim 30\%$ when a neighborhood tolerance of 150 km, in line with avoidance operations, is applied.

- When compared to the IFS forecast, the modified ARPEGE model exhibits a potential improvement in the representation of $RH_{ice}$ distribution around and beyond saturation. Regarding ISSR discrimination skills, the modified ARPEGE exhibits broadly similar capabilities to those of IFS, better for IFS when slightly subsaturated events are considered, and better for modified ARPEGE when high supersaturated events are considered. In both cases, the models have shortcomings in representing the highest values of $RH_{ice}$ (>105-110 % for IFS, >115 % for ARPEGE).

- Scores show that the operational ARPEGE, even without supersaturation physics, can also be used to discriminate ISSRs provided that a $RH_{ice}$ decision threshold lower than 100 % is applied. For IFS, using a $RH_{ice}$ threshold of 100 % to detect ISSR is shown to be sub-optimal and lower thresholds ( 98 %) seem to be more appropriate. This suggests that post-processing recalibration of $RH_{ice}$ outputs could be a viable solution when using NWP forecasts with non-allowing supersaturation schemes or more generally biased humidity outputs. However, it implies supplementary complexity for end-users. A model like the modified ARPEGE could allow for direct implementation without the need for adjustments, enabling flight trajectory optimization testing without additional effort.

- With regard to ISSR verification methodologies, the benefits of using spatial metrics are demonstrated. Indeed, in addition to solving traditional scoring issues like double penalty, spatial metrics are useful to relate the forecast skills at different spatial scales to the end-user application: here the avoidance margin applied to ISSR forecasts when rerouting flight paths. Sensitivity study is also conducted on humidity decision thresholds in order to comprehensively assess performance and make meaningful comparisons between discrimination skills of NWP models.

In the short term, the main perspectives will be to make the modified scheme operational, which involves: coupling the assimilation, analyzing the impact on the ARPEGE general circulation on a global scale, and readjusting the parameter set according to the results, if necessary. During this process, additional studies will be conducted to better understand the underlying reasons for the underrepresentation of the highest supersaturations and to explore the feasibility and value of including more detailed representation of cirrus cloud physics, such as water vapor deposition, in the ARPEGE microphysical framework. Additional checks may also be carried out using complementary observations (e.g. radiosondes) to study the performance of the models in capturing the vertical extent of ISSRs, which can be essential for avoidance strategies. As part of CICONIA SESAR Joint Undertaking (2023), with end-user partners, work on cost/loss metrics linking NWP model ISSR discrimination capabilities to traffic and climate metrics with associated $CO_2$ and non-$CO_2$ contributions is underway to explore the feasibility of avoidance scenarios.

## Appendix A: Modified Smith cloud scheme calculations

### A1 General expressions of cloud fraction and mean water content

To keep the framework simple, we assume that the nature of the probability distribution $G$ is independent of the value of $k$. This translates into the fact that the $q_t - k\,q_{sat}$ distributions $G_{[Q_{c,k},\sigma_{s,k}]}$ for different values of $k$ differ from each other by an offset and a scaling factor. The mean cloud fraction is given by the integration of $q_t - k q_{sat}$ distribution $G_{[Q_{c,k},\sigma_{s,k}]}$ for positive values only, which corresponds the cloudy part of the grid-box:

$$C^+ = \int_0^{+\infty} G_{[Q_{c,k},\sigma_{s,k}]}(q)\,dq. \tag{A1}$$

Introducing $t = (q - Q_{c,k})/\sigma_{s,k}$ scaling gives:

$$C^+ = \int_{-Q_{c,k}/\sigma_{s,k}}^{+\infty} G_{[0,1]}(t)\,dt, \tag{A2}$$

where $G_{[0,1]}$ is the centred and reduced probability distribution.

In a similar way, the mean condensate content after adjustment $\bar{q}_c^+$ is given by the expectation formula using the $q_t - q_{sat}$ distribution. The integration is proceeded on the cloudy part of the grid-box. In order to get a workable expression, it is assumed that $q_t - k\,q_{sat} > 0$ coincides with the highest values of $q_t - q_{sat}$. This allows to calculate $\bar{q}_c^+$ such that

$$\bar{q}_c^+ = \int_{q^*}^{+\infty} G_{[Q_{c,1},\sigma_{s,1}]}(q).q\,dq, \tag{A3}$$

where the lower bound of the interval $q^*$ is implicitly defined to ensure cloud fraction consistency with $C^+$ found in (A2) as shown in Fig. A1. This condition gives the following equality:

$$\int_{q^*}^{+\infty} G_{[Q_{c,1},\sigma_{s,1}]}(q)\,dq = \int_0^{+\infty} G_{[Q_{c,k},\sigma_{s,k}]}(q)\,dq = C^+. \tag{A4}$$

Re-scaling left and right members of equation (A4) by $t = (q - Q_{c,1})/\sigma_{s,1}$ (left) and $t = (q - Q_{c,k})/\sigma_{s,k}$ (right) leads to:

$$\int_{(q^*-Q_{c,1})/\sigma_{s,1}}^{+\infty} G_{[0,1]}(t)\,dt = \int_{-Q_{c,k}/\sigma_{s,k}}^{+\infty} G_{[0,1]}(t)\,dt \tag{A5}$$

which, by identification of lower integration bounds gives:

$$q^* = Q_{c,1} - Q_{c,k}\frac{\sigma_{s,1}}{\sigma_{s,k}}. \tag{A6}$$

This expression leads to the following formulation for the computation of the mean condensate content:

$$\bar{q}_c^+ = \int_{Q_{c,1}-Q_{c,k}\frac{\sigma_{s,1}}{\sigma_{s,k}}}^{+\infty} G_{[Q_{c,1},\sigma_{s,1}]}(q).q\,dq. \tag{A7}$$

Then, introducing $t = (q - Q_{c,1})/\sigma_{s,1}$ scaling gives the following expression for the mean cloud water content after adjustment:

$$645 \quad \bar{q}_c^+ = \sigma_{s,1} \int_{-Q_{c,k}/\sigma_{s,k}}^{+\infty} G_{[0,1]}(t) \left( t + \frac{Q_{c,1}}{\sigma_{s,1}} \right) dt. \tag{A8}$$

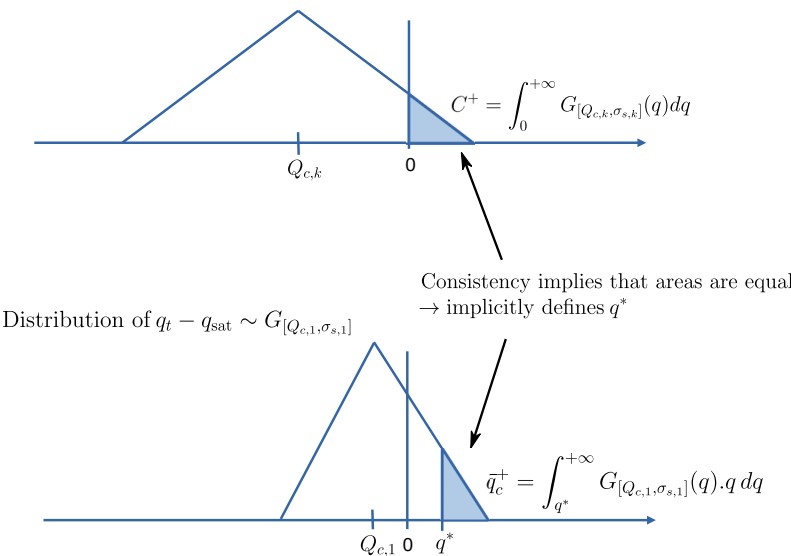

**Figure A1.** Illustration of integration steps to compute the mean cloud fraction $C^+$ and the mean condensate content $\bar{q}_c^+$ within the grid-box with triangular distributions. The first step is the integration of $q_t - kq_{sat}$ distribution on its positive support in order to compute cloudiness $C^+$. The second step is the integration of local cloud contents defined by the $q_t - q_{sat}$ distribution on the grid-box cloudy part, leading to $\bar{q}_c^+$. In this second step, the integration domain is $[q^*, +\infty[$ where $q^*$ is defined to ensure consistency on cloudiness computation.

## A2 Application with a triangular distribution

Symmetric triangular probability law is given by the following set of equations:

$$G_{[0,1]}(t) = \begin{cases} 0 & t \leq -\sqrt{6} \\ \frac{1}{6}t + \frac{1}{\sqrt{6}} & -\sqrt{6} < t \leq 0 \\ -\frac{1}{6}t + \frac{1}{\sqrt{6}} & 0 < t \leq \sqrt{6} \\ 0 & t > \sqrt{6} \end{cases} \tag{A9}$$

Easier calculations are obtained by introducing a re-scaled function $H(T) = \sqrt{6}G_{[0,1]}(\sqrt{6}T)$:

$$
\quad H(t) = \begin{cases} 0 & T \leq -1 \\ 1+T & -1 < T \leq 0 \\ 1-T & 0 < T \leq 1 \\ 0 & T > 1 \end{cases} \tag{A10}
$$

### A2.1 Computation of cloud fraction $C^+$

Integration of the cloud fraction can be expressed in the following manner:

$$
C^+ = \int_{-Q_{c,k}/\sigma_{s,k}}^{+\infty} G_{[0,1]}(t)\,dt = \int_{-\alpha_k\sqrt{6}}^{+\infty} G_{[0,1]}(t)\,dt = \int_{-\alpha_k}^{+\infty} \sqrt{6}\,G_{[0,1]}(\sqrt{6}T)\,dT = \int_{-\alpha_k}^{+\infty} H(T)\,dT. \tag{A11}
$$

**Case 1**: $\alpha_k \leq -1$

If $\alpha_k \leq -1$, the lower bound of integration is greater than 1 then $C^+ = 0$.

**Case 2**: $-1 < \alpha_k \leq 0$

If $-1 < \alpha_k \leq 0$, then $0 < -\alpha_k \leq 1$:

$$
C^+ = \int_{-\alpha_k}^{1} 1 - T\,dT = \frac{1}{2}(1+\alpha_k)^2. \tag{A12}
$$

**Case 3**: $0 < \alpha_k \leq 1$

If $0 < \alpha_k \leq 1$, then $-1 < -\alpha_k \leq 0$:

$$
C^+ = \int_{-\alpha_k}^{0} 1 + T\,dT + \int_{0}^{1} 1 - T\,dT = \int_{-\alpha_k}^{0} 1 + T\,dT + \frac{1}{2} = \frac{1}{2} - \frac{1}{2}(1-\alpha_k)^2 + \frac{1}{2} = 1 - \frac{1}{2}(1-\alpha_k)^2. \tag{A13}
$$

**Case 4**: $\alpha_k > -1$

In this case, $-\alpha_k < -1$, and result can be inferred from case 3:

$$
C^+ = \int_{-1}^{0} 1 + T\,dT + \int_{0}^{1} 1 - T\,dT = 1. \tag{A14}
$$

### 665 A2.2 Computation of mean condensed water $\bar{q}_c^+$

Integration of mean water content can be expressed in the following manner:

$$
\bar{q}_c^+ = \sigma_{s,1} \int_{-Q_{c,k}/\sigma_{s,k}}^{+\infty} G_{[0,1]}(t)\left(t + \frac{Q_{c,1}}{\sigma_{s,1}}\right)dt = \sigma_{s,1} \int_{-\alpha_k\sqrt{6}}^{+\infty} G_{[0,1]}(t)\left(t + \alpha_1\sqrt{6}\right)dt = \sigma_{s,1}\sqrt{6} \int_{-\alpha_k}^{+\infty} H(T)(\alpha_1 + T)\,dT. \tag{A15}
$$

**Case 1**: $\alpha_k \leq -1$

If $\alpha_k \leq -1$, the lower bound of integration is greater than 1 then $q_c^+ = 0$.

**Case 2**: $-1 < \alpha_k \leq 0$

If $-1 < \alpha_k \leq 0$, then $0 < -\alpha_k \leq 1$:

$$\bar{q}_c^+ = \sigma_{s,1}\sqrt{6} \int_{-\alpha_k}^{1} (1-T)\,(\alpha_1 + T)\,dT. \tag{A16}$$

Using integration by parts gives:

$$\bar{q}_c^+ = \frac{\sigma_{s,1}\sqrt{6}}{2} \int_{-\alpha_k}^{1} (1-T)^2\,dT - \frac{\sigma_{s,1}\sqrt{6}}{2}\left[(1-T)^2(\alpha_1+T)\right]_{-\alpha_k}^{1} \tag{A17}$$

$$= \frac{\sigma_{s,1}}{\sqrt{6}}(1+\alpha_k)^3 + \frac{\sigma_{s,1}\sqrt{6}}{2}\left[(1+\alpha_k)^2(\alpha_1-\alpha_k)\right]. \tag{A18}$$

**Case 3**: $0 < \alpha_k \leq 1$

If $0 < \alpha_k \leq 1$, then $-1 < -\alpha_k \leq 0$:

$$\bar{q}_c^+ = \sigma_{s,1}\sqrt{6} \int_{-\alpha_k}^{0} (1+T)\,(\alpha_1+T)\,dT + \sigma_{s,1}\sqrt{6} \int_{0}^{1} (1-T)\,(\alpha_1+T)\,dT. \tag{A19}$$

Applying integration by parts on the first integral and using results of case 2 for the second integral gives:

$$\bar{q}_c^+ = \frac{\sigma_{s,1}\sqrt{6}}{2} \int_{-\alpha_k}^{0} (1+T)^2\,dT - \frac{\sigma_{s,1}\sqrt{6}}{2}\left[(1+T)^2(\alpha_1+T)\right]_{-\alpha_k}^{0} + \left[\frac{\sigma_{s,1}}{\sqrt{6}} + \frac{\sigma_{s,1}\alpha_1\sqrt{6}}{2}\right] \tag{A20}$$

$$= -\frac{\sigma_{s,1}}{\sqrt{6}}\left[(1+T)^3\right]_{-\alpha_k}^{0} + \frac{\sigma_{s,1}\sqrt{6}}{2}\left[(1+T)^2(\alpha_1+T)\right]_{-\alpha_k}^{0} + \left[\frac{\sigma_{s,1}}{\sqrt{6}} + \frac{\sigma_{s,1}\alpha_1\sqrt{6}}{2}\right] \tag{A21}$$

$$= -\frac{\sigma_{s,1}}{\sqrt{6}} + \frac{\sigma_{s,1}}{\sqrt{6}}(1-\alpha_k)^3 + \frac{\sigma_{s,1}\alpha_1\sqrt{6}}{2} - \frac{\sigma_{s,1}\sqrt{6}}{2}(1-\alpha_k)^2(\alpha_1-\alpha_k) + \frac{\sigma_{s,1}}{\sqrt{6}} + \frac{\sigma_{s,1}\alpha_1\sqrt{6}}{2} \tag{A22}$$

$$= \sigma_{s,1}\alpha_1\sqrt{6} + \frac{\sigma_{s,1}}{\sqrt{6}}(1-\alpha_k)^3 - \frac{\sigma_{s,1}\sqrt{6}}{2}(1-\alpha_k)^2(\alpha_1-\alpha_k). \tag{A23}$$

**Case 4**: $\alpha_k > 1$

In this case, $-\alpha_k < -1$, then,

$$\bar{q}_c^+ = \sigma_{s,1}\sqrt{6} \int_{-1}^{0} (1+T)\,(\alpha_1+T)\,dT + \sigma_{s,1}\sqrt{6} \int_{0}^{1} (1-T)\,(\alpha_1+T)\,dT. \tag{A24}$$

Expression of $q_c^+$ can be inferred from case 3 by setting $\alpha_k = 1$. We obtain:

$$\bar{q}_c^+ = \sigma_{s,1}\alpha_1\sqrt{6}. \tag{A25}$$

## Appendix B:  The impact of the modifications in the cloud properties

In this section, we analyze how the modifications introduced in ARPEGE to allow supersaturation w.r.t ice affect the cloud properties, which in turn may influence the temperature trend. The diagnostics over the horizontal domain toolbox (DDH toolbox,  Piriou, 2025) allows us to quantify and analyze differences between the modified and non-modified models, ARP-new and ARP-op, for several ice cloud-related variables. The results are averaged by latitude bands for different pressure levels. As an illustrative case, we performed a diagnostic for the 6-hour forecast initialized on 1 January 2023 run 00UTC (Fig. B1).

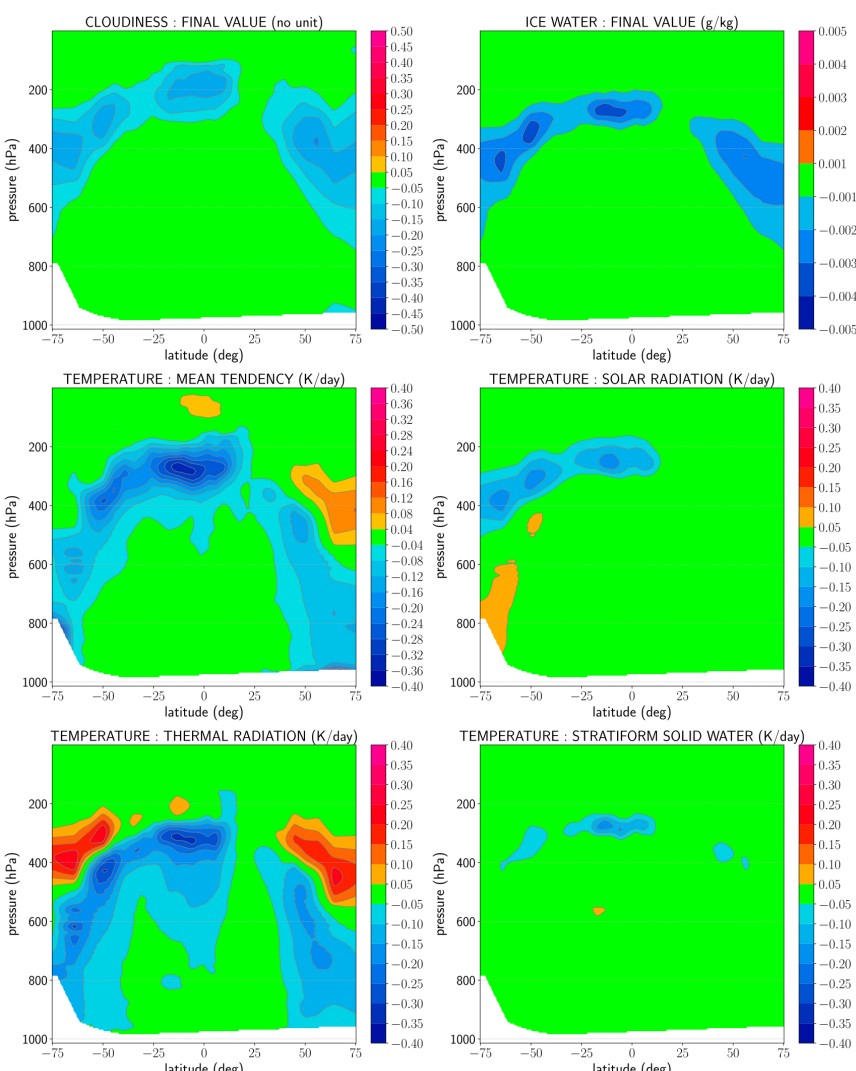

**Figure B1.** Differences between ARP-new and ARP-op for several parameters related to cloud processes at T+6h - 1 January 2023 run 00UTC. The results are averaged by latitude bands for different pressure levels. A positive or negative difference is highlighted by red (or blue) colours on the scale.

Figure B1 shows that the introduction of supersaturation leads to a significant decrease in cloudiness and ice content in the atmospheric layers close to the tropopause. The impact on temperature is mainly reflected in a general cooling at and below the layers affected by this modification. This difference in temperature trend is primarily attributable to three contributions: solar radiation, thermal radiation, and stratiform processes related to solid water, the thermal radiative contribution being dominant. Indeed, the solar radiation diagnostic shows that in regions where cloudiness decreases, more solar radiation is transmitted

downward, and less solar radiation is transmitted upward. The signal, however, remains weak comparatively to the thermal radiation diagnostic which reveals the opposite behavior of the dominant greenhouse role of cirrus clouds: with fewer cirrus, the trapping of outgoing terrestrial radiation is reduced, resulting in weaker longwave fluxes in the troposphere. At the same time, since less radiation is absorbed below, more longwave radiation escapes upward, which appears as an increase in fluxes at higher levels. Finally, a decrease in ice water content implies that less water vapor is undergoing condensation at high levels,

and therefore less latent heat is released, as shown in the stratiform solid water diagnostic.

. The Python code and the processed data that support the findings of this study and reproduce the main figures in this article are openly available at https://doi.org/10.5281/zenodo.15303979 (Arriolabengoa, 2025). The IAGOS data can be downloaded from the IAGOS data portal at https://doi.org/10.25326/06 (Boulanger et al., 2020). The forecast data used here are available upon reasonable request from the corresponding authors.

. SA, PC, OJ and YB developed the modification of the ARPEGE cloud scheme. YL and AP prepared and provided the IAGOS dataset. SA and PC developed the verification methodology. SA and OJ carried out the implementation of the new scheme in ARPEGE. SA, PC and MP took the lead in writing the manuscript. YB, MP and BV supervised the findings of this work. All authors provided critical feedback and helped shape the research, analysis and manuscript.

. No competing interests are declared.

. This research has received funding from the SESAR 3 Joint Undertaking under grant agreement No 101114613 (CICONIA) under European Union's Horizon Europe research and innovation programme. We gratefully acknowledge Daniel Cariolle and Maxime Perini from CERFACS, Toulouse, France and Valentin Curat from Météo-France for fruitful discussions. ECMWF is acknowledged as the HRES IFS data source. We acknowledge the use of imagery from the NASA Worldview application (https://worldview.earthdata.nasa.gov), part of the NASA Earth Science Data and Information System (ESDIS).

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
