# Peer review of "Modeling and verifying ice supersaturated regions in the ARPEGE model for persistent contrail forecast"

_EGUsphere, 2025_

## Author Comment (AC1)

**Reply to Anonymous Referee #2**

Referee comment on "Modeling and verifying ice supersaturated regions in the ARPEGE model for persistent contrail forecast" by S. Arriolabengoa et al. (egusphere-2025-1499, https://egusphere.co-pernicus.org/#RC2, 2025).

We thank the Referee for the time spent reviewing our manuscript and for the valuable comments, which have helped us to enhance the clarity of the paper and to provide a more thorough discussion of certain aspects. The answers to the various remarks are given as follows. For better legibility, the Referee's comments are highlighted with a gray background and changes in the manuscript are in italic.

**Review of the Manuscript**

The manuscript presents a modification to the ARPEGE NWP cloud scheme to allow for the representation of ice supersaturated regions (ISSRs), with evaluation against aircraft in situ humidity data and comparisons to the Integrated Forecast System (IFS). The topic is timely and highly relevant to aviation climate impact mitigation strategies. The paper is generally well-motivated and provides a solid contribution toward improved ISSR representation. However, there are several issues that need to be further clarified or elaborated before publication.

**Major comments**

1. While the study introduces a practical modification to ARPEGE's cloud scheme, the extent of its novelty compared to earlier approaches (e.g., in IFS or other global circulation models) is not entirely clear. The paper would benefit from a more explicit discussion of how the proposed approach differs from existing contrail cirrus cloud parameterizations used in current GCMs (e.g., ECHAM, CESM).

We agree with the general comment on the value of further discussion on how the proposed approach differs from existing parameterizations of condensation cirrus clouds. The first paragraph of Section 6.1 (L430) has been reworded and fully reorganized to include more precise remarks between similarities and differences concerning ARPEGE and IFS cloud schemes + other GCM models with ISSR representation capabilities.

A modified cloud scheme for ARPEGE NWP global model is presented in this work to enhance the representation of relative humidity w.r.t to ice, and in particular supersaturation, which is a necessary condition for contrail persistence. The modeling is based on a generalization of the Smith cloud scheme currently used in the operational ARPEGE. This modification can be implemented without any major modification and does not imply supplementary computational effort. A notable point is that the methodology developed in this article reworks the Sommeria and Deardorff (1977)

statistical concepts while incorporating ISSR parametrization, which allows for extensions to other atmospheric models using a similar framework. This is the case, for example, of the ICE3 microphysical scheme implemented in the French regional NWP model AROME (Seity et al., 2011, 2012), which uses a turbulence-based Gaussian distribution for the representation of clouds, and which could benefit from these developments.

Some common assumptions are shared with the parameterization implemented in the IFS model (Tompkins et al., 2007), in particular the use of a temperature function based on Koop et al. (2000) to describe the ice nucleation threshold, and the assumption that once this threshold is locally exceeded, local adjustment is instantly obtained back to saturation in the cloud. However, there are major differences between the two models in terms of their respective microphysical frameworks. For example, the cloud fraction is diagnostic in ARPEGE and prognostic in IFS. These different frameworks also imply differences in the representation of subgrid variability: IFS uses a dedicated distribution combining uniform (out-cloud) and Dirac (in-cloud) distributions, compatible with its prognostic cloud fraction parametrization, while ARPEGE uses a triangular distribution. The ARPEGE model also differs from other GCMs that allow supersaturation, such as ECHAM or CAM (see e.g. Bock and Burkhardt, 2016; Neale et al., 2010; Chen et al., 2012). These models are used for climate prediction and include supersaturation w.r.t ice, generally associated with a parametrization of contrail cirrus. In both cases, the triggering of homogeneous nucleation is proposed in a similar manner, mainly driven by a temperature function, but the thermodynamic in-cloud adjustment to saturation is relaxed, within a more complex framework of a 2-moment microphysical scheme.

2. The evaluation is primarily based on IAGOS aircraft measurements. While these are high-quality and relevant, the representativeness of IAGOS data (limited flight routes, sampling biases) should be more explicitly discussed. Could the conclusions change in regions less well sampled by IAGOS? The authors mention radiosondes as a possible next step, therefore an expanded discussion of current limitations in observational coverage would strengthen the evaluation framework. The authors should provide more detail on the dataset and discuss potential sampling biases.

Regarding the question of whether the conclusions could change in regions less well sampled by IA-GOS, the answer is addressed in the response to Special Comment 3. However, note that using data from a region of high data density, which coincides with a region of strong contrail cirrus, provides a solid foundation on which to evaluate NWP models.

Regarding the general comments on the observation datasets, the discussion on observations on L491 of Sect. 6.2 is reworded and enhanced to take into account the referee's demand:

In future works, other observational dataset could be useful for further verification of the NWP models. IAGOS dataset is open-accessible and quality-checked, relevant for humidity measurements in the UTLS. However, there is an heterogeneity in sampling due to the operational flight paths of participating airlines (Wolf et al., 2025). The highest measurement density is found across the North

Atlantic flight tracks, North-Eastern America and Europe at cruise altitudes (200hPa to 250hPa). In addition, the sampling is biased by avoiding aviation hazards (e.g. deep convective clouds and the outflow of such clouds, turbulent regions) and by route optimization according to the jet stream. Radiosonde observations which mainly sample the atmosphere above continental surfaces represent a valuable complementary database in world regions or weather situations where IAGOS data are less represented. This observation source also makes it possible to assess humidity at levels above or below typical cruising altitudes (see e.g. Bland et al., 2021; Thompson et al., 2024). Also, contrails tracked using satellite imagery [...]

3. The authors note that both ARPEGE and IFS underrepresent the highest values of RHice (> 110–115%). This is an important limitation, since persistent contrails are most strongly associated with highly supersaturated regions. Could the authors elaborate on the physical or numerical reasons why models fail at these extremes? Addressing this could help guide future model improvements.

Sperber and Gierens (2023) suspects that the cut-off in the histograms can be related to the assumption (commonly made in modified ARPEGE and IFS) that once the supersaturation threshold is locally exceeded, local adjustment is instantly obtained back to saturation, which leads that air inside cirrus clouds is assumed to be exactly at ice saturation. Indeed humidity grows progressively larger in the gridbox, and when locally crosses the supersaturation threshold, the local  $RH_{ice}$  is instantly dropped back to 100% while adding the excess vapor directly into solid phase. This is clearly an approximation as we know that in-cloud supersaturation can exist due the slow process of vapor deposition on pre-existing ice. This is also in line with the comment from referee 1 (specific point 3).

To clarify this point, we propose to rephrase the second paragraph from L442 in the Discussion Sect. 6.1:

For example, in the modified scheme, the adjustment in the cloud to ice saturation is assumed to be instantaneous, which is probably a major limitation of the physical description of ice representation in our context and an explanation of the cut-off on the highest supersaturation observed in the  $RH_{ice}$  distribution histogram (Fig. 6a). Indeed, it has been shown that  $RH_{ice}$  decreases with a relaxation time that can exceed several time steps, to finally reach a few percent above saturation, thus allowing local in-cloud supersaturation w.r.t ice. Sperber and Gierens (2023) proposed a cooling-cloud formation-phase relaxation process [...]

**and to add a comment in the perspectives L583:**

During this process [towards operationalization], additional studies will be conducted to better understand the underlying reasons for the underrepresentation of the highest supersaturations and to explore the feasibility and value of including more detailed representation of cirrus cloud physics, such as water vapour deposition, in the ARPEGE microphysical framework.

4. The modification to the cloud scheme is presented as computationally inexpensive and algorithmically simple, but the paper does not sufficiently explore whether introducing supersaturation impacts other parts of the model system. For example, how does it affect cloud microphysics, radiative transfer, or dynamics? Even if these effects are minor, they should be explicitly discussed.

Sentence L465 has been reworded to be more precise on the potential effects of introducing supersaturation on the model system and some figures have been added in the appendix to illustrate those effects, also in response to referee's special comment 6. The intensity of these impacts, in particular on the dynamics, and whether they benefit ARPEGE or not, cannot be assessed without a complete study, including the retroaction of observation assimilation which is beyond the scope of this paper. This work is currently underway with a view to implementing the changes operationally, and preliminary results show that after recalibrating some parameters of the microphysical scheme, a neutral to positive effect on prognostic variables such as temperature, wind or geopotential is obtained for the various regional domains of the WMO, traditionally used for NWP model verification.

**Modification of sentence L465:**

[...] Indeed, the introduction of supersaturation implies a reduction of the cloud cover and cloud ice contents in the altitudes which are impacted by the modification. This change will have a direct impact on temperature through changes in radiative and latent heat transfers, with an expected cooling effect, mainly in the upper troposphere (see Appendix B). The intensity of these impacts, their consequences for the dynamics, and whether they benefit or not for ARPEGE will be assessed in a dedicated forthcoming study.

**Special comments**

1. L102–103: What is the value of  $C_{calib}$  in this study? How do you get this value? Is  $C_{calib}$  expected to change with variations in time and geographic location?

 $C_{calib}$  is a global tuning coefficient attached to the ice nucleation threshold formulation. In our study, it is empirically obtained in Sect. 4 by comparing the forecast  $RH_{ice}$  distribution, with the observed  $RH_{ice}$  distribution with an independent dataset (see Fig. 5 with several values of  $C_{calib}$  empirically tested). In the paper this value is fixed but it could vary among different localizations (eg. tropical, mid-latitudes, high latitudes), or time. Note that for NWP models in an operational context, there is usually a tradeoff between complexity and accuracy to deal with, and changing the calibration across different variables could add substantial complexity.

To have a best understanding on how  $C_{calib}$  is defined, more details are given in sentence L220: The calibration is based on adjusting the saturation ratio coefficient k through the calibration coefficient  $C_{calib}$  (see Fig. 4). The value of  $C_{calib}$  is obtained empirically by comparing the predicted and observed distributions of  $RH_{ice}$  using a dedicated calibration dataset.

In the discussion on the impact of regional domains, we also added a reference to the section describing calibration to help the reader make the connection (see the answer to Special Comment 3).

2. L147–150: To what extent is the cloud fraction sensitive to the choice of probability distribution? What alternative approaches exist for representing probability distributions beyond the normalized centered probability distribution? What rationale did the authors provide for selecting the normalized centered probability distribution over other methods?

In statistical cloud schemes, the probability distribution represents subgrid variability through local distances to saturation within the gridbox. Since cloud cover is obtained by integrating local points above saturation, the characteristics of the distribution will have a direct impact on cloud cover. For example, assuming a pre-defined standard deviation, a triangular distribution will result in a non-zero cloud fraction for lower relative humidity values compared to a uniform distribution, but the increase in cloud fraction relative to the increase in mean relative humidity will be more gradual. This behaviour will be accentuated if a Gaussian distribution is used.

The distribution used in our work is Triangular (Smith scheme - see Fig. 2) and not Normal (I think there may be a misinterpretation here). The mention "normalized centered" in calculation steps L148 does not refer to a Gaussian distribution, and has to be interpreted as: "centered and reduced" which is a preferable formulation to avoid any confusion. We propose to modify this in the article. Note that the reason for the choice of a triangular distribution is due to the fact that the operational ARPEGE uses this formulation that we want to adapt to allow ISSR.

3. L184–185: This study excludes the United States and Asia. How could that influence the results?

North America and Asia are large continental surfaces with large mountain ranges and the repartition between processes leading to ISSR in the atmosphere (e.g. Warm Conveyor Belt, Convection) probably differs from the European / Northern Atlantic climatology, and could modify the results. However, subcontinental variability should be considered. For example, Petzold et al. (2020) shows that the climatology of H2O at cruising levels and ISSR occurrence in the Eastern North America (ENA) is rather similar to the North Atlantic and Europe. Regarding subtropical Asia, the ISSR occurrences are rather low because of higher temperature at cruising levels, which may induce adaptation of the models.

We completed sentence L461 by including North America and Asia in the discussion about the impact of selecting different domains. We also referenced the section on calibration to help readers make the connection:

However, we believe that this work could be extended to tropical regions. In this case, it could be calibrated accordingly, for example by modifying the  $C_{calib}$  coefficient (see the calibration procedure,

section 4), given that tropical climatology differs from mid-latitude regions and that the occurrence of ISSR depends on factors such as dominant deep convection. Similarly, North America and Asia are large continental areas with large mountain ranges, and the distribution of processes leading to ISSR in the atmosphere (e.g. warm convection belt, convection) is likely to differ from European/North Atlantic climatology, which could alter the results.

**4. L186: Please provide an explanation of FL250 and FL450.**

The sentence L186 has been reworded: The vertical domain extends from flight levels FL250 to FL450 (i.e. 375hPa to 150hPa), which correspond to altitudes of 25000ft and 45000ft in the ICAO standard atmosphere (ICAO, 1993), encompassing regions favorable to the triggering of persistent contrails.

**5. L215–218: Will the smoothing and undersampling alter the properties represented in the original IAGOS dataset?**

In response to this comment and to specific comment 2 from Referee 1, we have reworked Figure 6 in Section 5.1 by adding IAGOS unfiltered observations to the frequency histogram. As illustrated in Figure 6a, there is no significant difference between filtered and unfiltered histograms, showing that filtering does not alter the properties represented in the original IAGOS dataset.

Figure 6: (a) Frequency histogram of  $RH_{ice}$  (1 % bins) with the associated frequency bias on  $RH_{ice}$  > 100% and (b) Mean Absolute Error (MAE) computed for different categories of observed humidity (5 % bins). Results are shown for IAGOS observational dataset (blue), filtered and unfiltered, ARP-new (purple), ARP-op (green) and IFS (orange). (c) Distribution plot of model bias in  $RH_{ice}$ ,

computed against unfiltered IAGOS observations, for the three ARP-new, ARP-op and IFS models. Verification dataset from the 1st July 2022 to the 30th June 2023 within the aerial boundary of 80° W-40° E and 30-75° N, covering North Atlantic and Europe.

Further comments have also been added to discuss the new figure:

L259: Model results are compared with filtered and unfiltered IAGOS observations, the latter being used to compute the distribution plots of direct model errors (Fig. 6c).

L267: We note that there is no significant difference between filtered and unfiltered observation histograms, showing that applying a 100s mean-filter does not alter the properties represented in the original IAGOS dataset.

6. Section 6.1: Providing results that show how ARP-new alters the cloud properties compared to ARP-op would be very helpful.

We have added results showing how the ice water content and cloud fraction of ARP-op have changed in ARP-new. Diagnostics in the horizontal domain show how cloud properties change when supersaturation is allowed in the model, as well as how this changes affect the temperature and the radiation fluxes. These results have been included in the appendix after the calculations relating to the modifications made to the ARPEGE cloud scheme. Below are the figures and comments we have included:

In this section, we analyze how the modifications introduced in ARPEGE to allow supersaturation w.r.t ice affect the cloud properties, which in turn may influence the temperature trend. The diagnostics over the horizontal domain toolbox (DDH toolbox, Piriou, 2025) allows us to quantify and analyze differences between the modified and non-modified models, ARP-new and ARP-op, for several ice cloud-related variables. The results are averaged by latitude bands for different pressure levels. As an illustrative case, we performed a diagnostic for the 6-hour forecast initialized on 1 January 2023 run 00UTC (Fig. B1).

Figure B1 shows that the introduction of supersaturation leads to a significant decrease in cloudiness and ice content in the atmospheric layers close to the tropopause. The impact on temperature is mainly reflected in a general cooling at and below the layers affected by this modification. This difference in temperature trend is primarily attributable to three contributions: solar radiation, thermal radiation, and stratiform processes related to solid water, the thermal radiative contribution being dominant. Indeed, the solar radiation diagnostic shows that in regions where cloudiness decreases, more solar radiation is transmitted downward, and less solar radiation is transmitted upward. The signal, however, remains weak comparatively to the thermal radiation diagnostic which reveals the opposite behavior of the dominant greenhouse role of cirrus clouds: with fewer cirrus, the trapping of outgoing terrestrial radiation is reduced, resulting in weaker longwave fluxes in the troposphere. At the same time, since less radiation is absorbed below, more longwave radiation escapes upward, which appears as an increase in fluxes at higher levels. Finally, a decrease in ice water content implies that

less water vapor is undergoing condensation at high levels, and therefore less latent heat is released, as shown in the stratiform solid water diagnostic.

Figure B1: Differences between ARP-new and ARP-op betweenfor several parameters related to cloud processes at T+6h-1 January 2023 run 00UTC. The results are averaged by latitude bands for different pressure levels. A positive or negative difference is highlighted by red (or blue) colours on the scale.

7. L562: Why did the authors choose to emphasize results at a neighborhood tolerance of  $150 \,\mathrm{km}$ ?

We have chosen to focus on the results obtained with a neighbourhood tolerance of 150 km because this distance could be considered a reasonable margin allowing aircraft to avoid ISSR areas horizontally in an operational re-routing or alternative flight planning context. Note that this distance is also consistent with the work of Spichtinger and Leschner (2016), which shows, using the IAGOS dataset, that the decrease in  $RH_{ice}$  to significantly subsaturated values (considered as such when  $RH_{ice} < 70\%$ ) occurs approximately 100 km from the edges of ISSRs. We have reformulated the paragraph from L561 to make a clearer relation between spatial verification results and avoidance: Spatial verification allows to show that a good spatial agreement between forecast and observations is obtained. For example, discrimination capabilities when forecasting ISSRs with the modified ARPEGE at lead terms between 6 and 18 hours show a hit rate of  $\sim 80\%$  and a false alarm ratio of  $\sim 30\%$  when a neighbourhood tolerance of 150 km, in line with avoidance operations, is applied.

**Bibliography**

- Jake Bland, Suzanne Gray, John Methven, and Richard Forbes. Characterising extratropical near-tropopause analysis humidity biases and their radiative effects on temperature forecasts. *Quarterly Journal of the Royal Meteorological Society*, 147(741):3878–3898, 2021.
- Lisa Bock and Ulrike Burkhardt. The temporal evolution of a long-lived contrail cirrus cluster: Simulations with a global climate model. *Journal of Geophysical Research: Atmospheres*, 121(7): 3548–3565, 2016.
- Chih-Chieh Chen, Andrew Gettelman, Cheryl Craig, Patrick Minnis, and David P Duda. Global contrail coverage simulated by cam5 with the inventory of 2006 global aircraft emissions. *Journal of Advances in Modeling Earth Systems*, 4(2), 2012.
- ICAO ICAO. Manual of the icao standard atmosphere [: extended to 80 kilometres (262 500 feet). Third Edition, Technical Report Doc 7488-CD, 1993.
- Thomas Koop, Beiping Luo, Athanasios Tsias, and Thomas Peter. Water activity as the determinant for homogeneous ice nucleation in aqueous solutions. *Nature*, 406(6796):611–614, 2000.
- Richard B Neale, Chih-Chieh Chen, Andrew Gettelman, Peter H Lauritzen, Sungsu Park, David L Williamson, Andrew J Conley, Rolando Garcia, Doug Kinnison, Jean-Francois Lamarque, et al. Description of the near community atmosphere model (cam 5.0). NCAR Tech. Note Near/tn-486+STR, 1(1):1–12, 2010.
- Andreas Petzold, Patrick Neis, Mihal Rütimann, Susanne Rohs, Florian Berkes, Herman GJ Smit, Martina Krämer, Nicole Spelten, Peter Spichtinger, Philippe Nédélec, et al. Ice-supersaturated air masses in the northern mid-latitudes from regular in situ observations by passenger aircraft: vertical distribution, seasonality and tropospheric fingerprint. Atmospheric chemistry and physics, 20(13):8157–8179, 2020.
- Jean-Marcel Piriou. Ddh toolbox. https://github.com/UMR-CNRM/ddhtoolbox, 2025.
- Yann Seity, Pierre Brousseau, Sophie Malardel, G Hello, P Bénard, F Bouttier, Ch Lac, and V Masson. The arome-france convective-scale operational model. *Monthly Weather Review*, 139(3): 976–991, 2011.

- Yann Seity, Christine Lac, Francois Bouyssel, Sebastien Riette, and Yves Bouteloup. Cloud and microphysical schemes in arpege and arome models. In *Proceedings of the Workshop on Parametrization of Clouds and Precipitation (ECMWF)*, Reading, UK, pages 5–8, 2012.
- G Sommeria and James W Deardorff. Subgrid-scale condensation in models of nonprecipitating clouds. *Journal of the Atmospheric Sciences*, 34(2):344–355, 1977.
- Dario Sperber and Klaus Gierens. Towards a more reliable forecast of ice supersaturation: concept of a one-moment ice-cloud scheme that avoids saturation adjustment. *Atmospheric Chemistry and Physics*, 23(24):15609–15627, 2023.
- Peter Spichtinger and Martin Leschner. Horizontal scales of ice-supersaturated regions. Tellus B: Chemical and Physical Meteorology, 68(1):29020, 2016.
- Gregory Thompson, Chloé Scholzen, Scott O'Donoghue, Max Haughton, Roderic L Jones, Adam Durant, and Conor Farrington. On the fidelity of high-resolution numerical weather forecasts of contrail-favorable conditions. *Atmospheric Research*, 311:107663, 2024.
- Adrian M Tompkins, Klaus Gierens, and Gaby Rädel. Ice supersaturation in the ecmwf integrated forecast system. Quarterly Journal of the Royal Meteorological Society: A journal of the atmospheric sciences, applied meteorology and physical oceanography, 133(622):53–63, 2007.
- Kevin Wolf, Nicolas Bellouin, Olivier Boucher, Susanne Rohs, and Yun Li. Correction of era5 temperature and relative humidity biases by bivariate quantile mapping for contrail formation analysis. *Atmospheric Chemistry and Physics*, 25(1):157–181, 2025.

---

## Author Comment (AC2)

**Reply to Anonymous Referee #1**

Referee comment on "Modeling and verifying ice supersaturated regions in the ARPEGE model for persistent contrail forecast" by S. Arriolabengoa et al. (egusphere-2025-1499, https://egusphere.co-pernicus.org/#RC1, 2025).

We thank the Referee for the constructive feedback provided on our study. The insightful comments have helped us to improve the quality of the manuscript and have also suggested interesting directions for future research. The answers to the various remarks are given as follows. For better legibility, the Referee's comments are highlighted with a gray background and changes in the manuscript are in italic.

**General comments**

Overall I find the manuscript to be acceptable, although a few minor suggested revisions are mentioned below. It is clearly written and well organized with supporting evidence and logic and easy-to-follow outcomes. The main criticism (further discussed below) pertains to the persistent problem in some models of achieving the right outcome for the right reason.

**Specific points**

1. The discussion in Sections 2.2 and 2.3 could be made part of the appendix directly instead of including in the main body. The cloud scheme is already discussed in detail in the appendix so isn't it simple to keep all those details in one place? The fact that multiple closure methods were attempted could be omitted and only the one picked could be described. The rejected method seems impertinent to readers. During the research, the authors discovered a closure idea that was inferior but that happens frequently in model parameterization development. Which dead-end pathways to describe to readers is subjective, but it doesn't seem to add any insight directly to a physical problem being solved. As one manuscript reviewer's opinion only, I would not require this to be addressed in a revision, so the editor can decide if there is mutual agreement among reviewers.

Section 2.2 and Section 2.3 present the re-working of the cloud schemes concepts used in ARPEGE (Sommeria and Deardorff, 1977; Smith, 1990) when supersaturation is allowed. We included these sections in the main body because:

- 1- the paper is primarily focused on adapting the current ARPEGE model,
- 2- we see opportunities to apply the proposed methodology to other cloud schemes with similar statistical concepts,

so we would prefer that the structure of section 2 remain unchanged within the main body. In order to achieve a more straightforward flow in the main body, we suggest to simplify L144-153 to go directly to the cloud fraction and condensate mean content expressions, in sections 2.2 and leave more detailed comments in the appendix:

The mean cloud fraction  $C^+$  and mean cloud condensate content  $\bar{q}_c^+$  after adjustment can be expressed in relation to the centered and reduced probability distribution  $G_{[0,1]}(t)$  (see calculations in Appendix), such as

$$\begin{split} C^{+} &= \int_{-Q_{c,k}/\sigma_{s,k}}^{+\infty} G_{[0,1]}(t) \, dt, \\ \bar{q}_{c}^{+} &= \sigma_{s,1} \int_{-Q_{c,k}/\sigma_{s,k}}^{+\infty} G_{[0,1]}(t) \left( t + \frac{Q_{c,1}}{\sigma_{s,1}} \right) \, dt. \end{split}$$

About the remark on closures, we believe that including different closure methods could be useful when applying the proposed methodology for including supersaturation in the cloud scheme to a different atmospheric model. In that case, the calibration process will not yield the same results and the chosen closure may differ. However, it is true that this frequently happens when developing a model parametrization, so if the editor feels that omitting this part from the main body would make the text easier to read, we will modify it during the revision process.

2. While I agree that the scale of model data versus observations is extremely different, I believe it is insightful to see a distribution of the fundamental raw model data error. A good example is found in Fig. 5 of Thompson et al (2024). The frequency histograms of RHice in this manuscript's Fig. 6 provides a good indication of the changes in ARP-new vs. IFS and Obs, but a distribution plot of direct model error for every single IAGOS unfiltered observation is desired as well.

Figure 6 in Section 5.1 has been enhanced by adding a panel with the distribution plots required by the referee (Figure 6c) and also IAGOS unfiltered frequency histogram (Figure 6a) as required by referee 2. The legend has been modified accordingly.

Figure 6: (a) Frequency histogram of  $RH_{ice}$  (1 % bins) with the associated frequency bias on  $RH_{ice}$  > 100 % and (b) Mean Absolute Error (MAE) computed for different categories of observed humidity (5 % bins). Results are shown for IAGOS observational dataset (blue), filtered and unfiltered, ARP-new (purple), ARP-op (green) and IFS (orange). (c) Distribution plot of model bias in  $RH_{ice}$ , computed against unfiltered IAGOS observations, for the three ARP-new, ARP-op and IFS models. Verification dataset from the 1st July 2022 to the 30th June 2023 within the aerial boundary of 80° W-40° E and 30-75° N, covering North Atlantic and Europe.

Further comments have also been added to discuss the new figure:

L259: Model results are compared with filtered and unfiltered IAGOS observations, the latter being used to compute the distribution plots of direct model errors (Fig. 6c).

L267: We note that there is no significant difference between filtered and unfiltered observation histograms, showing that applying a 100s mean-filter does not alter the properties represented in the original IAGOS dataset.

L279: In the analysis of the direct model error, Fig. 6c shows that ARP-new has a median bias centered on 0, slightly better than the median bias of ARP-op and IFS. However, IFS shows a narrower interquartile range (11.06%), compared to (13.87%) for ARP-new. This indicates that, after bias correction, IFS has a slightly better overall accuracy in representing humidity.

3. Why are various models still not using a better physical representation of ice depositional growth from a physical means rather than using variants of saturation adjustment? Efforts to create and use tuning knobs to handle ice supersaturation rather than updating inherent physical growth equations seems endless. Eq. 1 is just another tuning knob component of three elements described in this paper: (1) a calibration coefficient; (2) a simplistic temperature function; and (3) a closure method that doesn't properly represent the physics as shown clearly in Fig. 5. The "cliff" in the histogram is related to Eq. 1 and the sentence in Line 112: "Once the supersaturation threshold is locally exceeded, local adjustment is instantly obtained back to saturation." In other words, as humidity grows progressively larger, it will cross the threshold and then suddenly the RHice is instantly dropped (let's say for example 145%) back to 100% while adding the excess vapor directly into solid phase.

There is no need to invoke a need for 2-moment cloud ice treatment to result in proper RHice forecasts. This appears to be a common misconception. A mass mixing ratio single moment scheme suffices with additional assumptions of ice spectral distribution. A basic inverse exponential distribution with a Y-intercept parameter that can increase as ice mass increases while holding a slope constant is one such assumption. This follows the most basic observations that more ice number comes with more ice mass. From whatever assumptions are made for number distribution, the total ice number (or number within bins of specific size ranges) can be diagnostically calculated, which effectively turns a 1-moment scheme into a 2-moment treatment. There is no solid evidence to say that 1-moment schemes are incapable of predicting the correct outcome compared to 2-moment schemes.

The essential problem of the microphysics is the lack of accounting for slow physical vapor depositional growth of ice. Creating a new threshold for when to convert instantly the excess vapor over ice saturation into cloud ice isn't solving the problem yet (as Fig. 5 clearly shows). In fact, the method to create initial ice where none previously existed could be fine with the new technique, but once ice does exist in a grid volume, do not permit more ice to nucleate and use a "electrical capacitance" analogy to grow the existing ice by vapor deposition. That way some of the excess (over saturation) water vapor can remain in gas phase and continue to permit  $RH_i > 100\%$ .

We thank the referee for sharing these comments and suggestions of improvement.

Our main objective was to improve the ISSR forecast ability of the operational global model ARPEGE. Modifying the microphysics parameterization of ARPEGE would have had large impacts on both forecast and assimilation systems. Our approach provides positive outputs that are much easier to propose for short-term operational use.

In the future, we plan to upgrade ARPEGE by using the physical parameterizations from our regional model AROME. Among these, the microphysics scheme would be ICE3 or LIMA. Explicit deposition on ice crystals is already available in LIMA (both with the 1-moment or 2-moment description of ice), and could be in ICE3 as well. Thus, it was not necessary at this time to work on the microphysics scheme of ARPEGE, but these ice-growing processes will be investigated and improved.

To be more specific about how to improve the system, we propose to rephrase the second paragraph from L442 in Sect. 6.1:

For example, in the modified scheme, the adjustment in the cloud to ice saturation is assumed to be instantaneous, which is probably a major limitation of the physical description of ice representation in our context and an explanation of the cut-off on the highest supersaturation observed in the  $RH_{ice}$  distribution histogram (Fig. 6a). Indeed, it has been shown that  $RH_{ice}$  decreases with a relaxation time that can exceed several time steps, to finally reach a few percent above saturation, thus allowing local in-cloud supersaturation w.r.t ice. Taking into account pre-existing ice and a better description of the physics of the vapor deposition should then improve the representation of the highest values of ISSR. Sperber and Gierens (2023) proposed [...]

With regard to the mention of 2-moment scheme, we suggest to modify L455-459 to add more precise comments:

Regarding the use of microphysical 2-moment schemes, we acknowledge that cloud formation processes can be better represented in NWP models involving more detailed physical processes, (Vié et al., 2016; Thompson et al., 2024; Seifert, 2024). For GCM climate models where contrail clouds are parametrized (e.g. Bock and Burkhardt, 2016; Chen et al., 2012), this type of scheme has proven useful for estimating the physical and optical properties of contrails, such as optical length (see e.g. Zhang et al., 2024). However, there is currently no consensus that obtaining a satisfactory representation of  $RH_{ice}$  in the UTLS necessarily requires the introduction of a second moment into the microphysical scheme, and the balance between computational costs and benefits in terms of improved forecasts must be carefully evaluated before implementing such a complex scheme in an operational NWP system.

**Technical corrections**

I did not exhaustively list many technical corrections because the manuscript was relatively good overall and I am late submitting the review so I am optimistic that other reviewers made more suggestions. Here are just a couple items.

L59: "verification methods deserve to be completed to accurately..." is awkward. It is simpler to state that verifying RHice in general is needed as well as threshold-based (ISSR) conditions?

The phrase has been rephrased as follows: Verification of  $RH_{ice}$  and threshold-based conditions is needed to accurately describe NWP model capabilities.

L60: "known to be a rather rare phenomenon in the atmosphere." It is not rare. It occurs 11% of the time in the entire atmosphere if you believe radiosonde data per Thompson et al (2024) or the manuscript's quote of 10% of the time from the IAGOS dataset. That does not seem especially rare. The phrase is basically repeated in L246.

It is true that, statistically, this is not a rare phenomenon and the wording could be misleading. We have changed the word and rephrased the two sentences as follows:

... known to be an occasional phenomenon in the atmosphere ... / ISSR occurrences are acknowledged to be an occasional phenomena in the atmosphere ...

**Bibliography**

- Lisa Bock and Ulrike Burkhardt. The temporal evolution of a long-lived contrail cirrus cluster: Simulations with a global climate model. *Journal of Geophysical Research: Atmospheres*, 121(7): 3548–3565, 2016.
- Chih-Chieh Chen, Andrew Gettelman, Cheryl Craig, Patrick Minnis, and David P Duda. Global contrail coverage simulated by cam5 with the inventory of 2006 global aircraft emissions. *Journal of Advances in Modeling Earth Systems*, 4(2), 2012.
- A. Seifert. Two-moment cloud ice scheme for predicting ice supersaturation in the global icon. Presented at the Contrail Analysis Workshop Google Research, London UK, 2024.
- RNB Smith. A scheme for predicting layer clouds and their water content in a general circulation model. Quarterly Journal of the Royal Meteorological Society, 116(492):435–460, 1990.
- G Sommeria and James W Deardorff. Subgrid-scale condensation in models of nonprecipitating clouds. *Journal of the Atmospheric Sciences*, 34(2):344–355, 1977.
- Dario Sperber and Klaus Gierens. Towards a more reliable forecast of ice supersaturation: concept of a one-moment ice-cloud scheme that avoids saturation adjustment. *Atmospheric Chemistry and Physics*, 23(24):15609–15627, 2023.
- Gregory Thompson, Chloé Scholzen, Scott O'Donoghue, Max Haughton, Roderic L Jones, Adam Durant, and Conor Farrington. On the fidelity of high-resolution numerical weather forecasts of contrail-favorable conditions. *Atmospheric Research*, 311:107663, 2024.
- Benoit Vié, J-P Pinty, Sarah Berthet, and Maud Leriche. Lima (v1. 0): A quasi two-moment microphysical scheme driven by a multimodal population of cloud condensation and ice freezing nuclei. *Geoscientific Model Development*, 9(2):567–586, 2016.
- Weiyu Zhang, Kwinten Van Weverberg, Cyril J Morcrette, Wuhu Feng, Kalli Furtado, Paul R Field, Chih-Chieh Chen, Andrew Gettelman, Piers M Forster, Daniel R Marsh, et al. Impact of host climate model on contrail cirrus effective radiative forcing estimates. *EGUsphere*, 2024:1–25, 2024.

---

## Author Response (AR2)

**Technical corrections before publication**

Response to the referee #2 and the editor's technical correction request for the manuscript "Modeling and verifying ice supersaturated regions in the ARPEGE model for persistent contrail forecast" by S. Arriolabengoa et al. (egusphere-2025-1499, https://doi.org/10.5194/egusphere-2025-1499, 2025).

We would like to thank the editor and referee #2 for their careful review. We appreciate their contribution to improving the quality of the paper.

A definition of the ARPEGE acronym was requested. We have addressed this correction in the revised version of the manuscript by including the acronym definition in both the abstract and the introduction (upon first appearance): ARPEGE (Action de Recherche Petite Echelle Grande Echelle).